# Representations and decodability of diverse cognitive functions are preserved across the human cortex, cerebellum, and subcortex

Tomoya Nakai [1,2✉] & Shinji Nishimoto[1,3,4]

Which part of the brain contributes to our complex cognitive processes? Studies have revealed contributions of the cerebellum and subcortex to higher-order cognitive functions; however, it has been unclear whether such functional representations are preserved across the cortex, cerebellum, and subcortex. In this study, we use functional magnetic resonance imaging data with 103 cognitive tasks and construct three voxel-wise encoding and decoding models independently using cortical, cerebellar, and subcortical voxels. Representational similarity analysis reveals that the structure of task representations is preserved across the three brain parts. Principal component analysis visualizes distinct organizations of abstract cognitive functions in each part of the cerebellum and subcortex. More than 90% of the cognitive tasks are decodable from the cerebellum and subcortical activities, even for the novel tasks not included in model training. Furthermore, we show that the cerebellum and subcortex have sufficient information to reconstruct activity in the cerebral cortex.

[1] Center for Information and Neural Networks, National Institute of Information and Communications Technology, Suita, Japan. [2] Lyon Neuroscience Research Center (CRNL), INSERM U1028 - CNRS UMR5292, University of Lyon, Bron, France. [3] Graduate School of Frontier Biosciences, Osaka University, Suita, Japan. [4] Graduate School of Medicine, Osaka University, Suita, Japan. ✉email: nakai.tomoya@neuro.mimoza.jp

Humans can perform various cognitive tasks, such as speech, calculation, memory retrieval, and decision-making. A central question of cognitive neuroscience is which part of the brain contributes to these higher-order cognitive functions. The cerebral cortex (or neocortex) has long been considered to play critical roles in human intelligence[1,2], whereas recent cross-species comparison across primates suggested that the expansion of cortical size or neuronal numbers is debatable[3,4]. In contrast to the cerebral cortex, the cerebellum contains four times the number of neurons[5]. During evolution, the cerebellar volume increased relative to the cerebral cortex in humans and other primates[6]. Cross-species comparison also revealed larger subcortical volumes, such as the hippocampus and amygdala, than in other primates[7]. Functional magnetic resonance imaging (fMRI) also demonstrated altered functional cortico-subcortical networks in humans compared with marmoset[8]. These previous studies suggest a possible contribution of functional reorganization of the cerebellum and subcortex to our higher-order cognitive functions.

Recent neuroimaging studies have shown the contribution of the cerebellum to multiple cognitive domains, such as motor coordination[9,10], language[11], emotion[12], working memory[13], and cognitive control[14], as well as partly overlapping networks of multiple cognitive functions[15–18]. Particularly, King et al. measured brain activity while subjects performed 47 task conditions and revealed functional parcellation comprising of 10 distinct functional subregions of the cerebellum[17].

Subcortex subregions are also associated with multiple cognitive abilities. For example, the hippocampus is involved in episodic memory[19,20], spatial navigation[21], learning and retrieval of sequential events[22]. The amygdala is involved in emotion recognition[23,24], uncertainty processing[25], and decision making[26]. Thalamus has been associated with multiple cognitive functions[27–29] and is considered to play the role of the "hub" in cortico-subcortical networks[30]. These previous studies have shown the contributions of the cerebellum and subcortex to higher-order cognitive functions, yet their representational organization has not been examined in a quantitative way.

Voxel-wise encoding models have been used to quantitatively evaluate various sensory features for their predictive performance of brain activity[31], as well as their modulation by selective attention[32,33]. Voxel-wise modeling also allows us to decode and reconstruct various visual images and movies from brain activity[34–36]. These methods have also been applied to higher-order brain functions, such as semantics[33,37–42] and emotion information[43,44]. Particularly, using both sparse task-type and continuous cognitive factor features, we visualized the cortical organization of 103 cognitive tasks in an earlier study and significantly decoded 95% of tasks based on the brain activity of the cerebral cortex[45]. However, it has not yet been examined how such representations differ among various brain parts, namely the cortex, cerebellum, and subcortex. To address these issues, we reanalyzed our previous fMRI data[45] and constructed voxel-wise encoding and decoding models independently using cortical, cerebellar, and subcortical voxels (Fig. 1). The current approach reveals representations of abstract cognitive functions not only in the cerebral cortex but also in the cerebellum and subcortex.

## Results

### Brain representations of task structures were preserved across the cortex, cerebellum, and subcortex.

To examine whether the representations of diverse cognitive tasks were similar across the cortex, cerebellum, and subcortex, we constructed a series of encoding models using sparse task-type features composed of one-hot vectors corresponding to the 103 tasks. Therefore, we visualized the representational structures of cognitive tasks using the representational similarity matrix (RSM) based on the weight matrices of the task-type encoding models for each region (Fig. 2a–c). The RSM was obtained by calculating the Pearson's correlation coefficients between the averaged weights of all task pairs across three time delays, concatenated across six subjects. The order of the 103 tasks in the RSMs were determined using hierarchical clustering analysis with the weight matrix of the cerebral cortex (Fig. 2a) and further applied to the orders of the other RSM of the cerebellum and subcortex (Fig. 2b, c, respectively). Overall, task structures were preserved across the cortex, cerebellum, and subcortex, which was quantified by positive correlation coefficients between elements of the RSM of the cortex and cerebellum (Spearman's correlation coefficient, $\rho = 0.861$; Fig. 2d) as well as between those of the cortex and subcortex ($\rho = 0.624$; Fig. 2e). Meanwhile, we also found a difference across the three brain parts. The standard deviation (SD) of representational task similarities was larger in the cortex than in the cerebellum and subcortex (Fig. 2f), suggesting that the structure of cognitive tasks is more distinctively organized in the cortex compared to the cerebellum and subcortex.

103 cognitive tasks varied in their visual and auditory inputs, and some tasks required motor outputs. The similarity of task structures across the three brain parts could be obtained merely by the difference of task-specific sensorimotor information. To exclude such a possibility, we first extracted visual features using the motion energy (ME) model, auditory features using the modulation transfer function (MTF) model, and motor features using the button response (BR) model (see **Motion energy features**, **Modulation transfer function features**, and **Button response features** subsections in Methods for detail) and concatenated those features to obtain sensorimotor features. We then performed the encoding model analyses 50 times within the training dataset using sensorimotor features and excluded the reliably predicted voxels (having a prediction accuracy of at least $r = 0.3$) from further analyses (sensorimotor voxels, Supplementary Fig. 1, Supplementary Table 1). RSMs were then obtained after excluding sensorimotor voxels. The task structures were not largely affected by this analysis (Supplementary Fig. 2). We again found significant correlations between the elements of the RSM of the cortex and cerebellum ($\rho = 0.832$) (Supplementary Fig. 2a) and between those of the cortex and subcortex ($\rho = 0.671$) (Supplementary Fig. 2b). We found similar results with the other thresholds ($r = 0.2, 0.1$) for selecting sensorimotor voxels (Supplementary Note 1). These results indicate that the abstract tasks structures were similar across the cortex, cerebellum, and subcortex.

### Metadata-based interpretation of the task organizations in the cerebellum and subcortex.

Although the similarity-based analyses in the previous section showed different representation patterns of 103 tasks in the cortex, cerebellum, and subcortex, it was unclear what cognitive factors contributed to these organizations. To interpret cognitive factors related to those tasks, we first performed principal component analysis (PCA) on the averaged weight matrix of the task-type model concatenated across six subjects. The resultant PCs were then associated with independent cognitive factors using a metadata-based reverse inference analysis. For each of the PC score maps, we calculated Pearson's correlation coefficients with the 715 reverse inference maps taken from the Neurosynth database[46]. The top and bottom 10 terms of each PC provided their objective interpretations (Supplementary Tables 2, 3). For the sake of intelligibility, we only presented the results of the top five PCs.

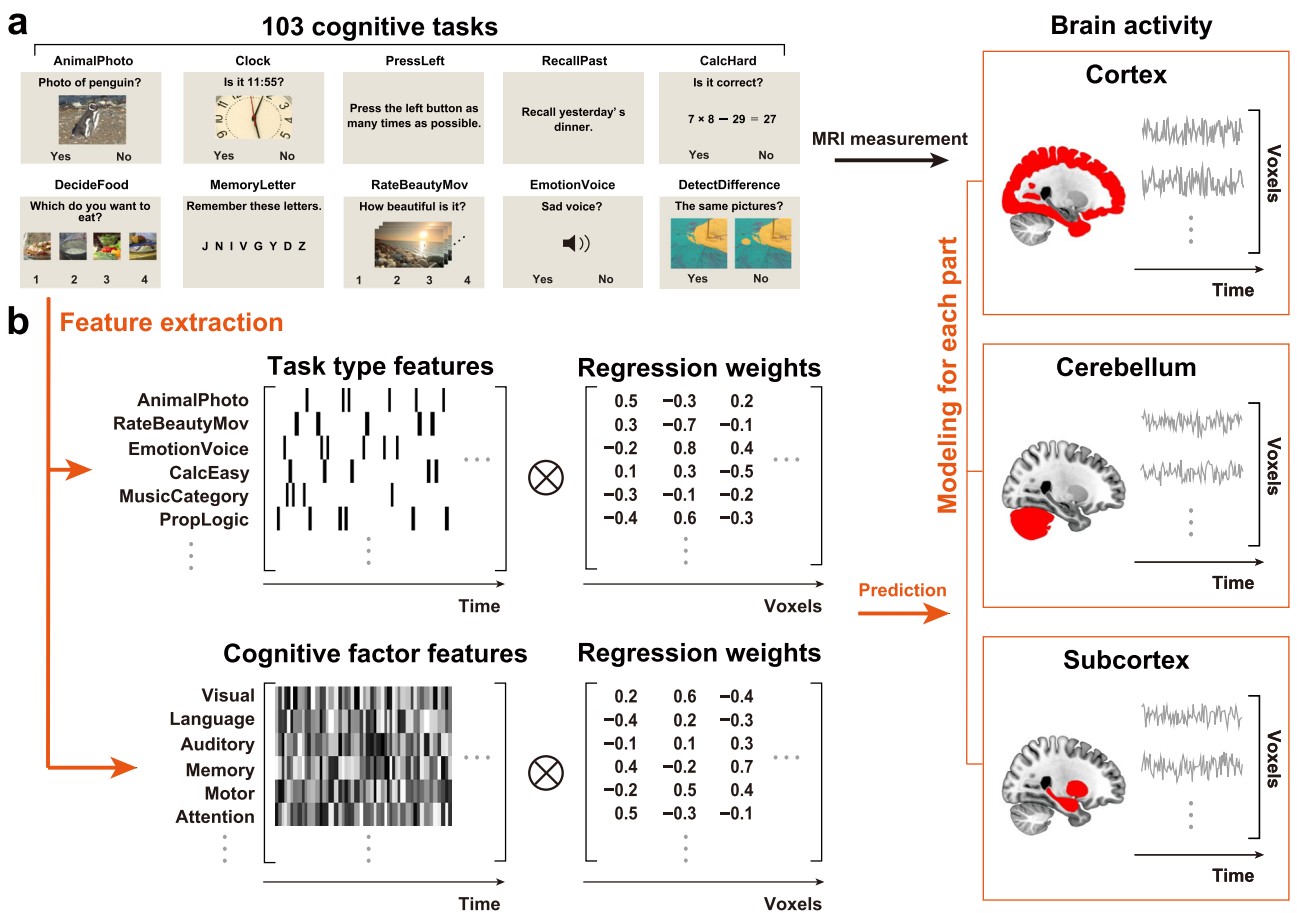

**Fig. 1 Experimental design and analyses.** Some figures were modified from one of our previous studies[45]. **a** The subjects performed 103 naturalistic tasks while the brain activity was measured using functional magnetic resonance imaging. Ten example tasks were shown. **b** Schematic of the encoding model fitting using the sparse task-type and continuous cognitive factor features. Three different encoding models were constructed using ridge regression, with cortical, cerebellar, and subcortical voxels.

For the cerebellum, PC1 was associated with introspection and emotion terms on the positive side (top 10 terms; "theory mind," "disgust") and executive function and motor terms on the negative side (bottom 10 terms; "working memory," "motor"). Contrarily, PC2 was associated with executive function terms on the positive side ("working memory," "execution") and introspection terms on the negative side ("autobiographical," "theory mind"). PC3 was associated with language terms on the positive side ("sentence," "comprehension") and motor terms on the negative side ("motor," "finger"). PC4 was associated with motor terms on the positive side ("finger," "sensorimotor") and language terms on the negative side ("reading," "linguistic"). PC5 was associated with executive function terms on the positive side ("cognitive task," "execution") and motor terms on the negative side ("finger," "motor").

For the subcortex, PC1 was associated with emotion terms on the positive side ("emotion," "valence"), whereas it was associated with motor terms on the negative side ("finger," "movement"). PC2 was associated with memory terms on the positive side ("memory," "retrieval") and somatosensory terms on the negative side ("pain," "somatosensory"). PC3 was associated with motor terms on the positive side ("muscle," "finger") and emotion terms on the negative side ("pain," "emotion"). PC4 was also associated with motor terms on the positive side ("preparation," "motor") and somatosensory terms on the negative side ("pain," "somatosensory"). PC5 was associated with memory terms on the positive side ("retrieval," "memory") and motor terms on the negative side ("finger," "sensorimotor").

**Visualization of representational structures of diverse tasks in the 2-dimensional cognitive spaces.** To provide a visual representation of diverse cognitive functions in different brain regions, we mapped all tasks onto 2-dimensional cognitive spaces using the loadings of the first and second PCs as the x-axis and y-axis, respectively (Cerebellum, Fig. 3a, Supplementary Fig. 3; Subcortex, Fig. 4a, Supplementary Fig. 4; see the visualization for the cortical voxels in our previous study[45]). The top five PCs explained 30.1% of the variances in the cerebellum weight matrix and 25.0% of the variances in the subcortex weight matrix (Figs. 3b, 4b). The 2D map based on the cerebellum showed consistent task organization with the metadata-based interpretation of PCs. Introspection tasks ("ImagineFuture", "RecallFace") are colored in red and located on the right side (i.e., on the positive side of the PC1). Tasks related to the executive function ("CalcHard", "PropLogic") are colored in green and located on top (i.e., on the positive side of the PC2). Language tasks ("Sarcasm", "WordMeaning") are colored in blue. The 2D map obtained based on the subcortex also showed consistent task organization with the metadata-based interpretation of PCs. Emotional tasks ("RateHappyPic", "RateDisgustPic") are colored in red and located on the right side (i.e., on the positive side of the PC1). Memory tasks ("LetterFluency", "RecallFace") are colored in green and located on top (i.e., on the positive side of the PC2). Motor tasks ("PressOrdHard", "PressLeft") are colored in blue.

To scrutinize representational differences in each subregion of the cerebellum and subcortex, we visualized average weight values of 103 tasks in each subregion (Figs. 3c–d, 4c–d). Based on the

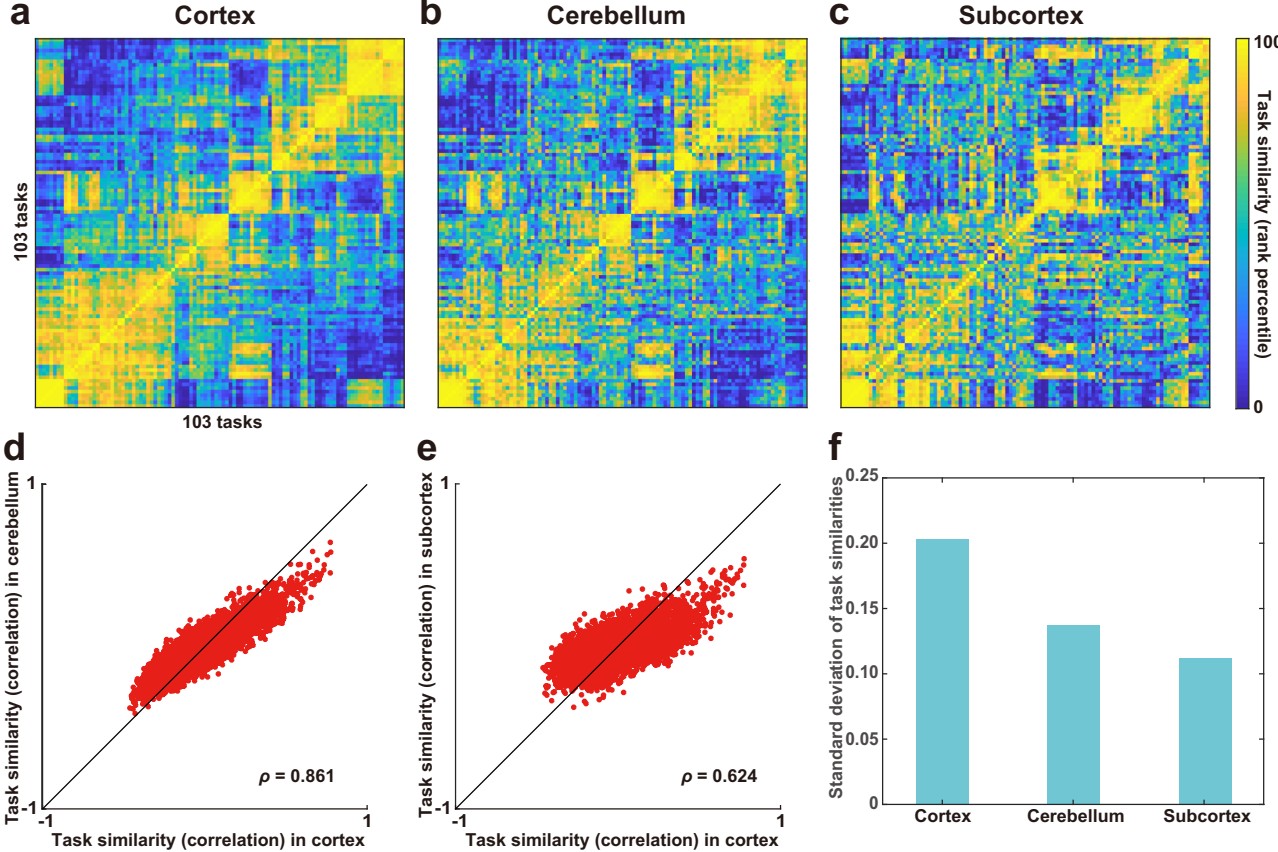

**Fig. 2 Similar task representations across the cortex, cerebellum, and subcortex. a–c** The representational similarity matrix (RSM) was constructed based on task similarity (Pearson's correlation coefficient) between each task pairs for the **a** cortex, **b** cerebellum, and **c** subcortex. Task similarity was calculated using weight vectors of all voxels included in the target region, concatenated across six subjects. A total of 103 tasks were ordered based on hierarchical clustering analysis for the cortical data. Task similarity was transformed to the rank percentile for the purpose of visualization. Task similarities were plotted **d** for the cortex and cerebellum, and **e** for the cortex and subcortex ($n = 5,253$). **f** Standard deviation of task similarities across the cortex, cerebellum, and subcortex.

previous multiple task study[17], we used function-based multi-domain task battery (MDTB) parcellations of the cerebellum. For example, motor movement tasks such as "Press Right" had a positive weight in MDTB_2 regions of interests (ROI) (labeled as "Right-hand presses," Fig. 3c). "Language" tasks had a positive weight in MDTB_8 ROI (labeled as "Word Meaning," Fig. 3d) (see the results of other subregions in Supplementary Fig. 5). These functional associations were consistent with functional parcellation labels[17], indicating that functional organization in the cerebellum was well captured by the visualization method of the current study.

A similar analysis was also performed for the subcortex subregions. For example, memory and imagery tasks had a positive weight in the left hippocampus (Fig. 4c), whereas demanding tasks, such as calculation, had a positive weight in the right caudate (Fig. 4d) (see the results of other subregions in Supplementary Fig. 6). These results showed how representations of multiple cognitive functions are distributed in the subcortex.

**Decoding of novel cognitive tasks from the activity of the cortex, cerebellum, and subcortex.** To examine the specificity of how multiple cognitive tasks were represented in the different parts of the brain, we constructed decoding models for each, the cerebral cortex, cerebellum, and subcortex (Fig. 5a). To further assess the generalizability of the decoding models to novel tasks, we extracted 715 latent cognitive factors related to each of the 103 tasks and constructed cognitive factor feature matrices.

Cognitive factor features were calculated based on Pearson's correlation coefficients between the weight maps of the task-type model and reverse inference maps of the Neurosynth[46] (see the **Cognitive factor features** subsection in Methods for details). We then trained a decoding model to estimate the cognitive factor features for 80% of the tasks and decoded the remaining 20%.

The decoding model of the cerebral cortex significantly decoded more than 95% of the cognitive tasks (decoding accuracy, mean ± SD, 0.952 ± 0.009; Fig. 5b top, Supplementary Tables 4–5) (see Supplementary Fig. 7 for the data of the other subjects). Significance of the decoded accuracy was further evaluated using a one-sided sign test; more than 99% of tasks were significantly decoded (mean ± SD, 99.5% ± 0.5% of the tasks were significant; $P < 0.05$, false discovery rate [FDR]-corrected). We also found that more than 90% of the cognitive tasks were significantly decoded using only cerebellar voxels (decoding accuracy, mean ± SD, 0.918 ± 0.015; 97.1% ± 1.9% of the tasks were significant; Fig. 5b middle) or subcortical voxels (decoding accuracy, mean ± SD, 0.856 ± 0.013; 92.7% ± 2.2% of the tasks were significant; Fig. 5b bottom), although the overall decoding accuracies were smaller than that of the decoding model based on the cortical voxels. To check the robustness of our decoding results, we performed permutation tests by randomly shuffling task labels in the test dataset for a total of 5,000 times. This analysis showed that tasks were significantly decoded in all three brain regions for all subjects ($P < 0.001$).

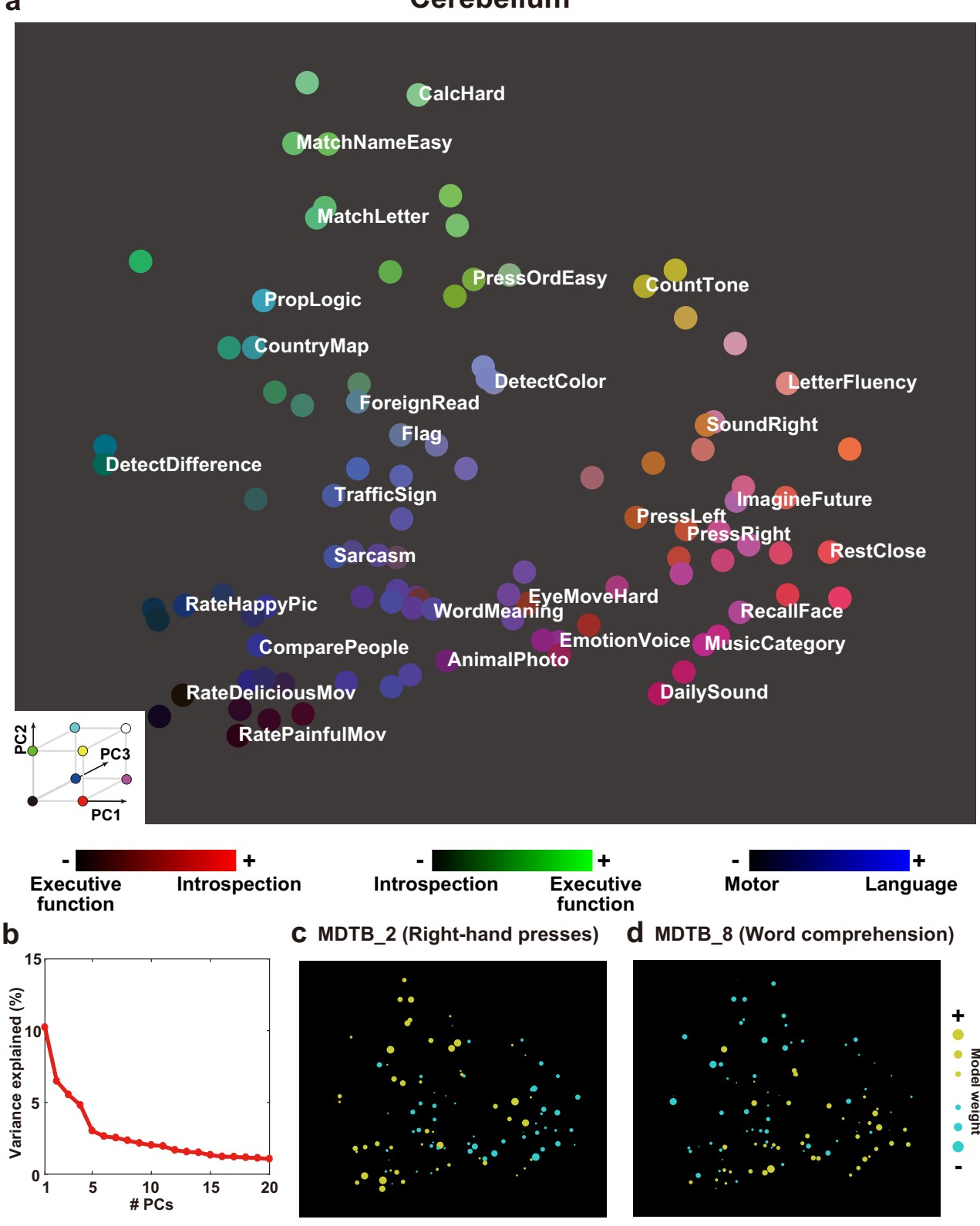

**a** Cerebellum

CalcHard
MatchNameEasy
MatchLetter
PropLogic
CountryMap
ForeignRead
Flag
DetectDifference
TrafficSign
Sarcasm
RateHappyPic
ComparePeople
RateDeliciousMov
RatePainfulMov
PressOrdEasy
CountTone
DetectColor
LetterFluency
SoundRight
ImagineFuture
PressLeft
PressRight
RestClose
EyeMoveHard
WordMeaning
RecallFace
EmotionVoice
MusicCategory
AnimalPhoto
DailySound

PC2 / PC3 / PC1

- Executive function  + Introspection
- Introspection  + Executive function
- Motor  + Language

**b**

Variance explained (%) vs # PCs

**c** MDTB_2 (Right-hand presses)

**d** MDTB_8 (Word comprehension)

+ Model weight −

The above one-vs.-one method provided very high decoding accuracy for all three brain parts; thus, the variability of decoding performances across tasks was unclear because of this ceiling effect. To examine such variability, we quantified decoding accuracy using task score (see Methods for details). We found that 99.5% ± 0.5% of tasks were significantly decoded using cortical activity (decoding accuracy, mean ± SD, 0.592 ± 0.018; Fig. 5c top, Supplementary Tables 6–7), 98.7% ± 0.5% of tasks were significantly decoded using cerebellar activity (decoding accuracy, 0.520 ± 0.027; Fig. 5c middle), and 95.5% ± 2.3% of tasks were significantly decoded using subcortical activity (decoding accuracy, 0.404 ± 0.020; Fig. 5c

**5**

**Fig. 3 Visualization of task structures in the cerebellum. a** Color and spatial visualization of the cognitive space. Colors indicate the normalized loadings of the top three principal components [PC1, red; PC2, green; PC3, blue] of the task-type model weights (concatenated across subjects), mapped onto the two-dimensional cognitive space based on the loadings of PC1 and PC2. Each PC is labeled based on metadata-based interpretation analysis (Supplementary Table 2). All tasks are presented in white letters. For a better visibility, only 30 tasks are shown in white. **b** Variance explained in the PCA. The explained variance of the original weight matrix of the task-type model was plotted for each PC. **c, d** Examples of task selectivity for voxels in the functional subregions in the cerebellum (multi-domain task battery (MDTB) parcellation[17]) of **c** MDTB_2 (Right-hand presses), and **d** MDTB_8 (Word comprehension), mapped onto the same two-dimensional cognitive space as **a** for subject ID01. Tasks with positive and negative weight values in **c** and **d** were indicated in yellow and cyan, respectively. The circle size was modulated based on the absolute weight value.

bottom) (See Supplementary Fig. 7 for the data of the other subjects). Permutation tests also showed that tasks were significantly decoded in all three brain regions for all subjects ($P < 0.001$).

To further examine whether the observed decoding accuracy is affected by low-level sensorimotor factors, we constructed a decoding model without sensorimotor voxels (see Methods for details). The model decoded most cognitive tasks from cortical (decoding accuracy, mean ± SD, 0.950 ± 0.008; 99.2% ± 0.7% of the tasks were significant), cerebellar (decoding accuracy, 0.915 ± 0.015; 96.8% ± 2.1% of the tasks were significant), and subcortical activities (decoding accuracy, 0.856 ± 0.012; 92.4% ± 2.1% of the tasks were significant) (Supplementary Fig. 8, Supplementary Tables 4–5). These results indicated that the cortex, cerebellum, and subcortex have abstract representations of cognitive functions, which can distinguish diverse cognitive tasks.

Decoding performances were similar between the cortex and cerebellum (Spearman's correlation coefficient, mean ± SD, $\rho = 0.808 ± 0.020$; Fig. 5d), as well as between the cortex and subcortex ($\rho = 0.627 ± 0.060$; Fig. 5e, Supplementary Fig. 9). Positive correlations of decoding performances were again found after excluding the sensorimotor voxels (between the cortex and cerebellum, $\rho = 0.838 ± 0.025$; cortex and subcortex, $\rho = 0.671 ± 0.081$, Supplementary Fig. 10), indicating that even the cerebellar and subcortical voxels can cover a sufficient portion of our cognitive space to be generalized to novel tasks.

To test decoding performance in a model-independent way, we also decoded over 100 tasks directly from brain activity using a support vector machine (Supplementary Fig. 11, Supplementary Tables 8–9). Note that this analysis did not decode novel tasks. We discovered that most tasks were significantly decoded from cortical (decoding accuracy, mean ± SD, 0.975 ± 0.021; all tasks were significant; one-sided sign tests, $P < 0.05$, FDR-corrected), cerebellar (decoding accuracy, 0.875 ± 0.006; 98.7% ± 3.2% of the tasks were significant), and subcortical activities (decoding accuracy, 0.756 ± 0.037; 95.0% ± 4.3% of the tasks were significant).

**Reconstruction of cortical activity from cerebellar and subcortical activities.** Finally, we tested whether the activity in the cerebral cortex can be reconstructed from activities in the cerebellum and subcortex (Fig. 6a). We first applied PCA to the brain activity of the cerebellum and subcortex and reduced it to 2,000 dimensions. This analysis preserved 89.2% ± 1.7% (mean ± SD) and 93.8% ± 1.2% of variances of the activity in the cerebellum and subcortex, respectively. We thus used these activities as feature matrices in encoding models to predict cortical activity. The cerebellum encoding model significantly predicted the activity of 95.7% ± 1.6% of cortical voxels (prediction accuracy, 0.385 ± 0.022; Fig. 6b, Supplementary Fig. 12). The subcortex encoding model significantly predicted the activity of 94.1% ± 2.7% of cortical voxels (prediction accuracy, 0.320 ± 0.025; Fig. 6c, Supplementary Fig. 13). The cerebellum + subcortex encoding model significantly predicted the activity of 96.3% ± 2.3% of cortical voxels (prediction accuracy, 0.392 ± 0.033; Figs. 6d, e,

Supplementary Fig. 14). The prediction performances of the above three models were relatively high compared with the encoding models using sparse task-type features (0.296 ± 0.055; 82.6% ± 7.2% of the cortical voxels were significant) and cognitive factor features (0.337 ± 0.049; 87.9% ± 5.2% of the cortical voxels were significant) (Fig. 6f).

To exclude the possibility that the prediction performances of cerebellum and subcortex models are caused by correlated noise among three brain parts, we constructed additional encoding models after subtracting average brain response across the entire brain in each time point. These models again predicted activity in most cortical regions (prediction accuracy of the cerebellum model 0.365 ± 0.020; the subcortex model, 0.295 ± 0.016; the cerebellum + subcortex model, 0.365 ± 0.029). These results indicate that a large part of the cortical activity can be reconstructed based on the cerebellar and subcortical activities.

## Discussion

In the current study, we examined representations of abstract cognitive functions stored in the cerebellum and subcortex using a voxel-wise modeling approach. Representational similarity analysis (RSA) revealed similar task structures across the cortex, cerebellum, and subcortex. By using cerebellar and subcortical activities, we decoded a large portion of cognitive tasks, including novel tasks that were not included in the model training. Encoding model analysis further supports our findings that the cerebellum and subcortex contain sufficient information to reconstruct cortical activity.

In our previous study of voxel-wise modeling of cognitive functions, we focused only on the activity of the cerebral cortex[45]. In the current study, we extended our previous approach to the cerebellum and subcortex, which contrasted with most previous voxel-wise modeling studies focused on the activity in the cerebral cortex. One previous study examined the cerebellum's contribution to multiple linguistic information using a voxel-wise modeling approach, which revealed a distribution of higher-order semantic concepts in the cerebellum[47]. Another voxel-wise modeling study used movie stimuli and reported emotion representations distributed in the subcortical regions and cerebellum[43]. In addition to the linguistic and emotional information, the current study also covered a wide variety of cognitive domains using a larger number of tasks compared with previous studies[15–18], providing a powerful tool to comprehensively compare functional organizations between the cortex, cerebellum, and subcortex.

Although the RSM showed a smaller SD in the cerebellum and subcortex than in the cortex, such difference in task similarity might be caused by the low signal-to-noise ratio (SNR) in the cerebellum and subcortex, rather than by the intrinsic distinctiveness of task representations. It is possible that task representations are equally distinct in the cerebellum and subcortex but are less clear due to the signal/noise quality of the current fMRI measurement. Further improvement in the measurement of cerebellar and subcortical activity is needed to disentangle the

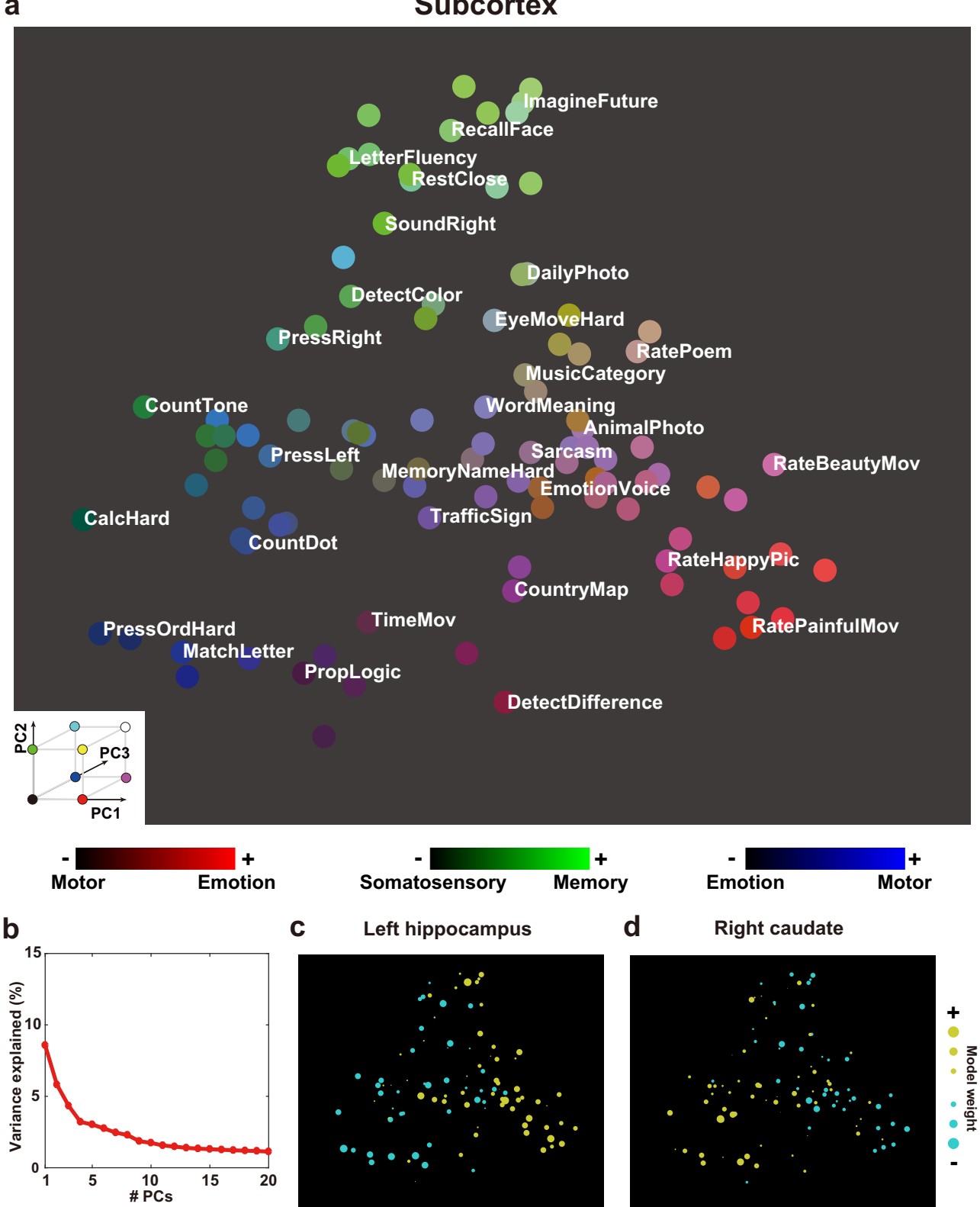

**Fig. 4 Visualization of task structures in the subcortex. a** Color and spatial visualization of the cognitive space. Colors indicate the normalized loadings of the top three principal components [PC1, red; PC2, green; PC3, blue] of the task-type model weights (concatenated across subjects), mapped onto the two-dimensional cognitive space based on the loadings of PC1 and PC2. Each PC is labeled based on metadata-based interpretation analysis (Supplementary Table 3). All tasks are presented in white letters. For a better visibility, only 30 tasks are shown in white. **b** Variance explained in the PCA. The explained variance of the original weight matrix of the task-type model was plotted for each PC. **c–d** Examples of task selectivity for subcortical voxels in the **c** left hippocampus and **d** right caudate, mapped onto the same two-dimensional cognitive space as **a** for subject ID01. Tasks with positive and negative weight values in **c** and **d** were indicated in yellow and cyan, respectively. The circle size was modulated based on the absolute weight value.

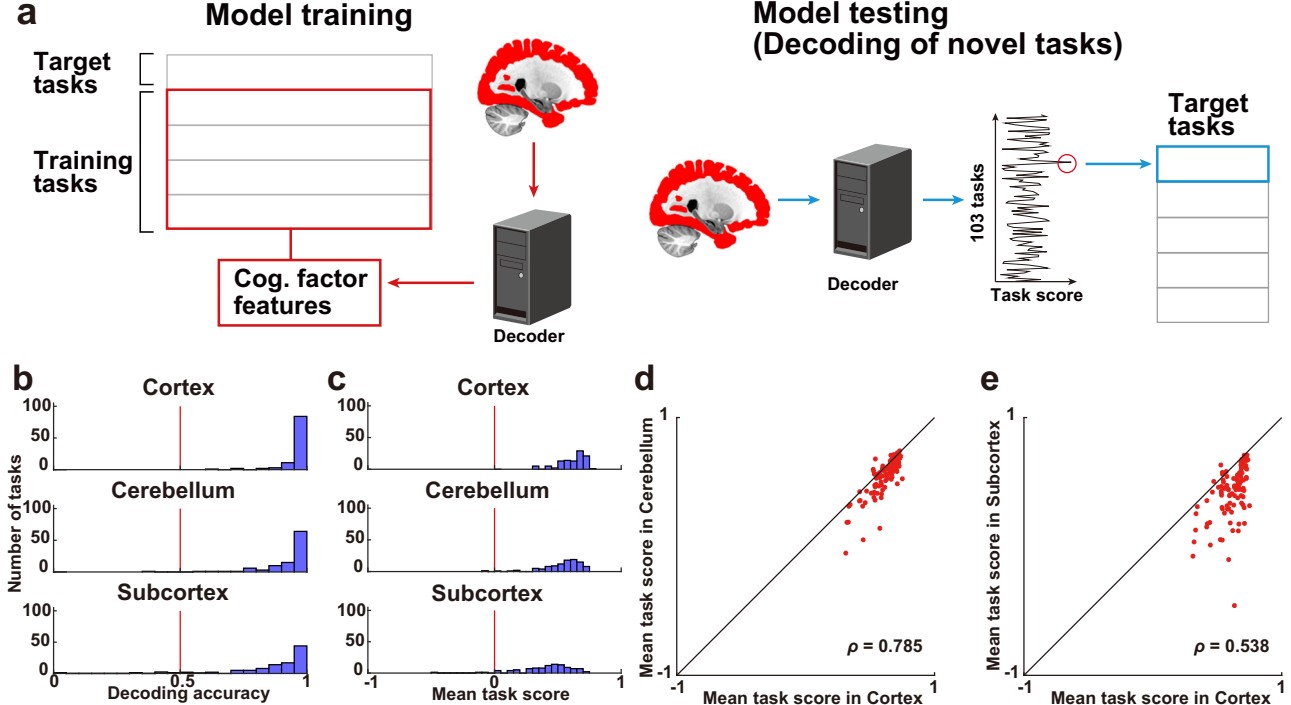

**Fig. 5 Decoding novel tasks. a** The decoding model was constructed using 80% of the tasks in the training dataset and applied to the 20% left-out tasks in the test dataset using brain activity with hemodynamic temporal delays. The decoding accuracy was examined by calculating task score for each target time point. **b** Histogram of task decoding accuracies for subject ID01, using a binary classification (Supplementary Fig. 7 show the other subjects). The red line indicates the chance-level accuracy (0.5). Filled bars indicate tasks that were decoded with significant accuracy (one-sided sign test, $n = 102$, $P < 0.05$, FDR-corrected). **c** Histogram of task decoding accuracies represented by average task score ($n = 715$). The red line indicates the chance-level accuracy (0). **d**–**e** Scatter plot of the task score of decoded tasks for 103 tasks by **d** the cortex and cerebellum models and **e** the cortex and subcortex models ($n = 103$).

SNR effect from the distinctiveness of task representations across different brain regions.

The metadata-based inference analysis revealed that both positive and negative sides of the top five PCs were associated with introspection/emotion, executive function, language, and motor terms. The involvement of the cerebellum in these cognitive factors has been reported in various previous studies (e.g., motor[9,10], language[11,47], emotion/introspection[12,48], and executive function[14,49]). We further investigated the functional contributions of cerebellum subregions to these cognitive factors using the functional parcellation of King et al. (2019)[17]. We adopted functional ROIs (fROIs) instead of anatomical ROIs because the study reported the dissociation between anatomical and functional parcellation. In line with this, fROIs have functional labels, which would be appropriate for testing the validity of the current study. Consistent with functional labels, we found that motor tasks such as "PressRight" had a larger weight than "PressLeft" in the cerebellar subregion MDTB_2 (Right-hand presses), whereas "PressLeft" had a larger weight in the MDTB_1 (Left-hand presses) (Fig. 3c, Supplementary Fig. 5a). Language-related tasks such as "WordMeaning" and "MoralPersonal" had larger weights on the positive side of PC3 (colored blue in Fig. 3a, related to language terms) in the MDTB_7 (Narratives) and MDTB_8 (Word comprehension) (Fig. 3d, Supplementary Fig. 5f). Demanding tasks such as "CalcHard" and "RelationLogic" had larger weights on the negative side of the PC1 and positive side of the PC2 (related to executive function terms) in the MDTB-5 and MDTB-6 (Divided attention) (Supplementary Fig. 5d, 5e). Imagination and recall tasks such as "RecallFace" and "ImagineFuture" had larger weights on the positive side of PC1 (related to

introspection terms) in the MDTB_10 (Autobiographical recall) (Supplementary Fig. 5h). These results confirmed the validity of functional parcellation in the cerebellum and demonstrated the diversity of task representations even within the same fROIs.

As for the subcortex, both positive and negative sides of the top five PCs were associated with emotion, memory, and motor terms, whereas the negative sides were further associated with somatosensory terms. The association of these terms was likely due to the contribution of subcortex subregions. The bilateral amygdala had larger weights on the positive side of PC1 (related to emotion terms; Supplementary Fig. 6c, d), which is consistent with previous studies reporting involvement of this region in emotion recognition[23,24]. The bilateral hippocampus was more weighted on the positive side of PC2 (related to emotion and memory terms; Fig. 4d and Supplementary Fig. 6b), which was consistent with previous studies in episodic memory and retrieval[19,20,22]. The bilateral caudate was more weighted on the negative side of PC1 (related to motor terms; Fig. 4d and Supplementary Fig. 6b), in line with previous studies of motor control and learning[50,51]. The bilateral thalamus was more weighted on the negative side of PC2 (related to somatosensory terms; Fig. 4d and Supplementary Fig. 6b), which is consistent with the widely-known view of this region as a pathway of sensory information[52,53]. Note that we used anatomical ROIs in the analysis of subcortex subregions. Although a recent study provided a detailed parcellation based on functional connectivity gradients[54], we did not adopt this parcellation because the functional labels were not provided for this atlas. Further investigation may clarify distinct cognitive spaces within each subcortical structure using such parcellation.

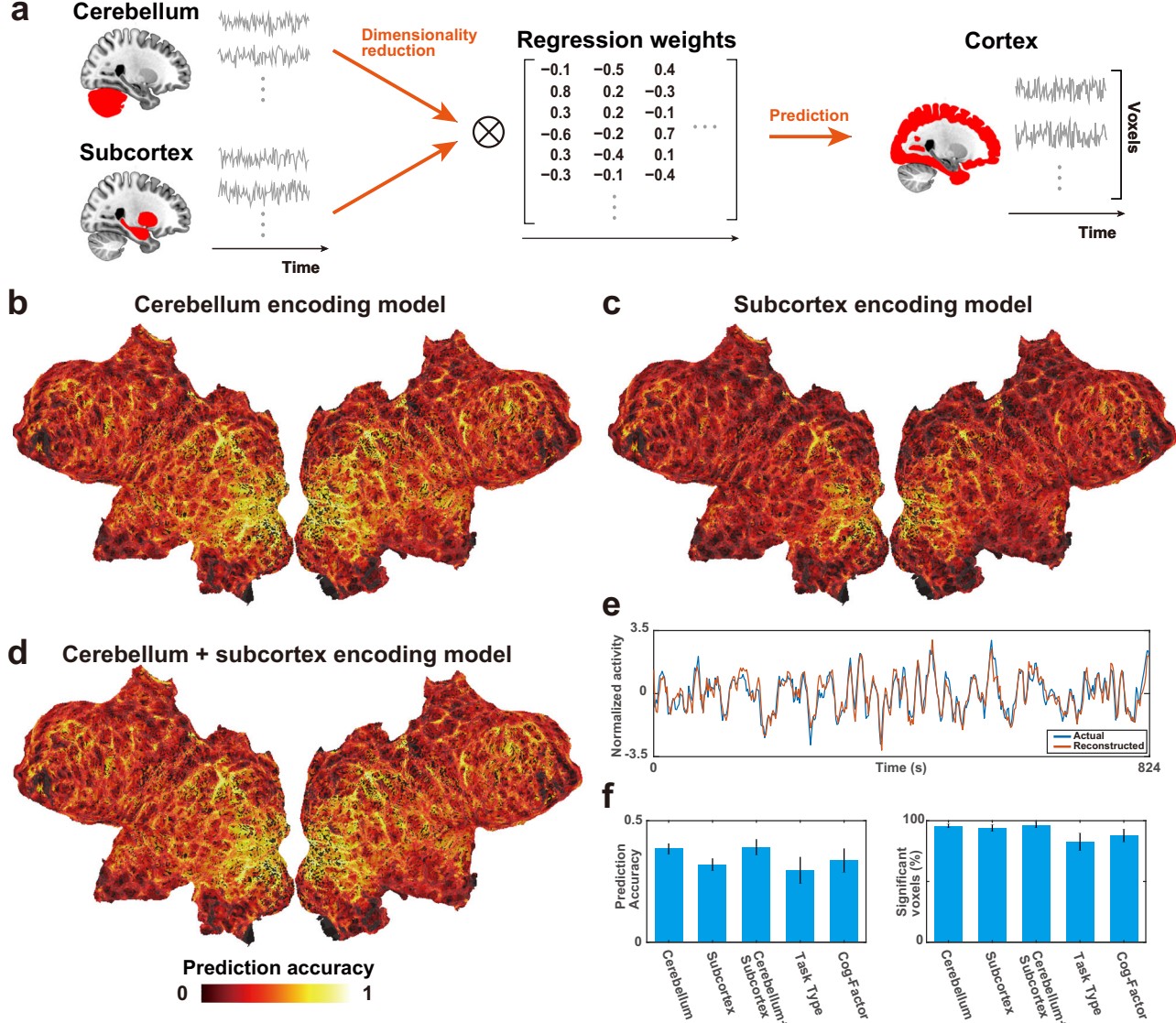

**Fig. 6 Reconstruction of cortical activity from cerebellar and subcortical activities. a** Reconstruction analysis of cortical activity from cerebellar and subcortical activities. Principal component analysis was applied to the original response matrices of the cerebellum and subcortex to reduce their dimensions to 2,000. These matrices were then used as feature matrices to predict cortical activity. **b**–**d** Cortical map of model prediction accuracy using the **b** cerebellum, **c** subcortex, and **d** cerebellum + subcortex encoding models, mapped on the flattened cortical sheets of subject ID01 ($n = 412$, $P < 0.05$, FDR-corrected; dashed line indicates the threshold). **e** Normalized actual (blue) and reconstructed (red) brain activity in the example cortical voxel of subject ID01, based on the cerebellum + subcortex encoding model. **f** Bar plots of mean prediction accuracy (left) and percent significant voxels (right) of cerebellum, subcortex, cerebellum + subcortex, task-type, and cognitive factor encoding models. Error bar, SD (calculated across subjects, $n = 6$).

To exclude the possibility that similar task representations were caused simply by low-level sensorimotor components, we performed additional analysis by excluding voxels that were predictable by the sensorimotor features. This analysis was more conservative than the analysis adopted in our previous study, which concatenated sensorimotor features with the target features[45] or quantitatively assessed the ratio of explained variances[33,39]. This is because the current approach excludes voxels which may contain both sensorimotor and task-related information. Although the current approach may underestimate representational similarity and decodability across the three brain parts, it more robustly ruled out the possible contribution of sensorimotor features. The current approach also had another advantage that allowed us to perform decoding analysis after excluding sensorimotor voxels, in contrast to the previous

approach that was applicable only for encoding models. We thus concluded that the high decoding performance of the cortex, cerebellum, and subcortex was not caused by the sensorimotor components of the multiple tasks.

In order to reconstruct the cerebral activity from the cerebellar and subcortical activities, we adopted the voxel-to-voxel encoding modeling technique[55–57]. The voxel-to-voxel encoding models use brain activity as input instead of stimulus-induced features and can capture unexplained variances by the latter features[57]. These models outperformed encoding models using task-type or cognitive factor features, indicating that the cerebellum and subcortex shared sufficient information of cognitive functions with the cerebral cortex, which was not fully captured by stimulus-induced features. The voxel-to-voxel modeling paved the way for the construction of appropriate features to represent human cognitive functions. Future

works are necessary to clarify in more detail functional similarity and distinctions across the cortex, cerebellum, and subcortex.

## Methods

**Subjects**. As stated in our previous study[45], six healthy subjects (aged 22–33 years, 2 females & 4 males; referred to as ID01–ID06) with normal vision and hearing participated in the current experiment. All subjects were right-handed (laterality quotient = 70–100), as assessed using the Edinburgh inventory[58]. Written informed consent was obtained from all subjects prior to their participation in the study. This experiment was approved by the ethics and safety committee of the National Institute of Information and Communications Technology in Osaka, Japan.

**Stimuli and procedure**. As explained in the earlier study[45], we prepared 103 naturalistic tasks that could be performed without any pre-experimental training (see Supplementary Note 2 for the detailed description of each task). Tasks were selected to include as many cognitive domains as possible. Each task had 12 instances; 8 instances were used in the training runs, whereas 4 were used in the test runs. The stimuli were presented on a projector screen inside the scanner (21.0 × 15.8° of visual angle at 30 Hz). The root-mean square of the auditory stimuli was normalized. During scanning, subjects wore MR-compatible ear tips. The experiment was performed for 3 days, with six runs performed each day. Presentation software (Neurobehavioral Systems, Albany, CA, USA) was used to control the stimulus presentation and collection of behavioral data. To measure button responses, optic response pads with two buttons in each of the left and right hands were used (HHSC-2×2, Current Designs, Philadelphia, PA, USA).

The experiment consisted of 18 runs, with 12 training runs and 6 test runs. Each run contained 77–83 trials with a duration of 6–12 s per trial. To keep subjects attentive and engaged and to ensure all runs had the same length, a 2-s feedback for the preceding task (correct or incorrect) was presented 9–13 times per run. In addition to the task, 6 s of imaging without a task was inserted at the beginning and at the end of each run; the former was discarded in the analysis. The duration of a single run was 556 s. In the training runs, task order was pseudorandomized, as some tasks depended on each other and were therefore presented close to each other in time (e.g., the tasks "MemoryDigit" and "MatchDigit"). In the test runs, 103 tasks were presented four times in the same order across all six runs (but with different instances for each repetition). There was no overlap between the instances in the training runs and test runs. No explanation of the tasks was given to the subjects prior to the experiment. During the fMRI experiment, subjects were instructed on how to perform each task by the instruction text that was shown as a part of the stimuli. Subjects only underwent a short training session on how to use the buttons used to respond.

**MRI data acquisition**. The experiment was conducted using a 3.0 T scanner (TIM Trio; Siemens, Erlangen, Germany) with a 32-channel head coil. We scanned 72 interleaved axial slices that were 2.0-mm thick without a gap, parallel to the anterior and posterior commissure line, using a T2*-weighted gradient echo multiband echo-planar imaging sequence [repetition time (TR) = 2,000 ms, echo time (TE) = 30 ms, flip angle (FA) = 62°, field of view (FOV) = 192 × 192 mm², resolution = 2 × 2 mm², MB factor = 3]. We obtained 275 volumes for each run, with each following three dummy images. For anatomical reference, high-resolution T1-weighted images of the whole brain were also acquired from all subjects with a magnetization-prepared rapid acquisition gradient echo sequence (MPRAGE, TR = 2,530 ms, TE = 3.26 ms, FA = 9°, FOV = 256 × 256 mm², voxel size = 1 × 1 × 1 mm³).

**fMRI data preprocessing**. Motion correction in each run was performed using the statistical parametric mapping toolbox (SPM12; Wellcome Trust Centre for Neuroimaging, London, UK; http://www.fil.ion.ucl.ac.uk/spm/). All volumes were aligned to the first echo planar image for each subject. Low-frequency drift was removed using a median filter with a 120-s window. The slice timing correction was performed to the first slice of each scan. The response for each voxel was then normalized by subtracting the mean response and scaling it to the unit variance. We used FreeSurfer[59,60] to identify the cortical surfaces from the anatomical data and register them to the voxels of the functional data. For each subject, the voxels identified in the cerebral cortex (53,345–66,695 voxels per subject), cerebellum (12,505–15,262 voxels per subject), and subcortex were used in the analysis (5,384–6,622 voxels per subject). For the subcortex, voxels in the bilateral hippocampus, caudate, amygdala, accumbens, pallidum, putamen, and thalamus were included. Note that we refined preprocessing parameters (e.g., using SPM12 instead of SPM8 and adding slice timing correction) compared with our previous study[45], which resulted in a slightly different distribution of representational similarities and decoding accuracies in the cerebral cortex.

**Task-type features**. The task-type features were composed of one-hot vectors, which were assigned 1 or 0 for each time bin, indicating whether one of the 103 tasks was performed in that period. The total number of task-type features was thus 103.

**Encoding model fitting**. In the encoding model, cortical activity in each voxel was fitted with a finite impulse response model that captured the slow hemodynamic response and its coupling with neural activity[35,61]. The feature matrix $\mathbf{F_E}$ [T × 3 N] was modeled by concatenating sets of [T × N] feature matrices with three temporal delays of 2, 4, and 6 s (T = # of samples; N = # of features). The cortical response $\mathbf{R_E}$ [T × V] was then modeled by multiplying the feature matrix $\mathbf{F_E}$ with the weight matrix $\mathbf{W_E}$ [3 N × V] (V = # of voxels):

$$\hat{\mathbf{R}}_\mathbf{E} = \mathbf{F_E}\mathbf{W_E} \qquad (1)$$

We used an L2-regularized linear regression using the training dataset to obtain the weight matrix $\mathbf{W_E}$. The training dataset consisted of 3,336 samples (6,672 s). The optimal regularization parameter was assessed using 10-fold cross-validation, with the 18 different regularization parameters ranging from 1 to $2^{17}$.

The test dataset consisted of 412 samples (824 s, repeated four times). To reshape the data spanning over six test runs into the four times-repeated dataset, we discarded 6 s of the no-task period at the end of each run as well as the 2-s feedback periods at the end of the third and sixth test runs. Four repetitions of the test dataset were averaged to increase the signal-to-noise ratio. Prediction accuracy was calculated using Pearson's correlation coefficient between the predicted signal and measured signal in the test dataset.

**Evaluation of optimal regularization parameters**. To keep the scale of the weight values consistent across subjects, we performed a resampling procedure to assess the optimal regularization parameter used for group RSA and PCA[38]. To this end, we randomly divided the training dataset into training samples (80%) and validation samples (20%) for each subject and performed model fitting using an L2-regularized linear regression. This procedure was repeated 50 times. The resultant prediction accuracies were averaged across the six subjects for each parameter. We selected the optimal regularization parameter that provided the highest mean prediction accuracy across subjects. This regularization parameter was used for model fitting for group RSA and PCA.

**Representational similarity analysis**. To examine hierarchical relations across tasks, we conducted an RSA. First, we concatenated the weight matrix of predictive voxels of the task-type model across six subjects. Concatenation of the estimated weights was performed to obtain a group-level representation that provides a common basis that is comparable across subjects[37,38]. To choose predictive voxels, for each subject, we selected the voxels that exhibited a significant prediction accuracy with $P < 0.05$ (with FDR correction) and averaged three time delays for each task. We then obtained the RSM by calculating the Pearson's correlation coefficients between the averaged weights of all task pairs. A dendrogram of the 103 tasks was then obtained using the task dissimilarity (1 - correlation coefficient) as a distance metric, using the minimum distance as a linkage criterion. For the purpose of visualization, tasks were reordered based on the dendrogram obtained using the cortical data.

**Principal component analysis of task-type weights**. For each brain region (cortex, cerebellum, or subcortex), we performed PCA on the weight matrix of the task-type model concatenated across six subjects. We selected the voxels that showed significant prediction accuracy with $P < 0.05$ (with FDR correction) and averaged three time delays for each task. To show the structure of the cognitive space, 103 tasks were mapped onto the two-dimensional space using the loadings of PC1 (first PC) and PC2 as the x-axis and y-axis, respectively. The tasks were further colored in red, green, and blue based on the relative PCA loadings in PC1, PC2, and PC3, respectively.

To represent the cortical organization of the cognitive space for each subject, we extracted and normalized the PCA scores from each subject's voxels. The resultant cortical map indicated the relative contribution of each cortical voxel to the target PC (denoted as *PCA score map*).

To obtain an objective interpretation of the PCs, we performed metadata-based inference of the cognitive factors related to each PC. We used Neurosynth as a metadata reference of the past neuroimaging literature[46]. From the approximately 3,000 terms in the database, we manually selected 715 terms that covered the comprehensive cognitive factors while also avoiding redundancy. In particular, we removed several plural terms that also had their singular counterpart (e.g., "concept" and "concepts") and past tense verbs that also had their present counterpart (e.g., "judge" and "judged") from the dataset. We also excluded those terms that indicated anatomical regions (e.g., "parietal"). We used the reverse inference map of the Neurosynth database for each of the 715 selected terms. The reverse inference map indicated the likelihood of a given term being used in a study if the activity was observed at a particular voxel. Each reverse inference map in the MNI152 space was then registered to the subjects' reference echo planar imaging (EPI) data using FreeSurfer[59,60]. For each of the PCA score maps, we calculated Pearson's correlation coefficients with the 715 registered reverse inference maps, which resulted in a cognitive factor vector with 715 elements. Terms with higher correlation coefficient values were regarded as contributing more to the target PC.

**Cognitive factor features**. To obtain task representations using continuous features in the human cognitive space, we transformed sparse task-type features into

latent cognitive factor feature space. We used the reverse inference map of the Neurosynth database[46] for each of the 715 terms selected. Each reverse inference map in the Neurosynth database in MNI152 space was registered to the subjects' reference EPI data using FreeSurfer[59,60].

We then calculated the Pearson's correlation coefficients between the weight map for each task in the task-type model and the registered reverse inference maps. This resulted in the [103 × 715] coefficient matrix. We next obtained the cognitive transform function (CTF) for each subject by averaging the coefficient matrices of the other five subjects. The CTF served to transform the feature values of the 103 tasks into the 715-dimensional latent feature space. The feature matrix of the cognitive factor model was then obtained by multiplying the CTF with the feature matrix of the task-type model. Note that the CTF (and the resultant feature matrix) of each target subject was independent of their own data. The total number of cognitive factor features was 715.

**Exclusion of sensorimotor voxels**. To exclude the possible effect of low-level sensorimotor features on prediction and decoding performances, we performed an additional encoding model fitting using sensorimotor components. Sensorimotor features were obtained by concatenating ME features (visual), MTF features (auditory), and BR features (motor) (see the following subsections for details). Our previous studies validated the efficacy of these feature in controlling sensorimotor information[33,44,45]. Moreover, ME and MTF features are easier to interpret than neural network features because they explicitly modeled neuronal response patterns in the early visual and auditory cortices. For each subject, we randomly divided the training dataset into training (80%) and validation samples (20%) and performed model fitting using an L2-regularized linear regression. This procedure was repeated 50 times. The reliably predicted voxels by this analysis (having a mean prediction accuracy of at least 0.3) were called as *sensorimotor voxels* and were excluded from some of the analyses as described in the Result section.

**Motion energy features**. We used the ME model that has been used in previous studies[35,44,45] and provided in a public repository (https://github.com/gallantlab/motion_energy_matlab). First, movie frames and pictures were spatially downsampled to 96 × 96 pixels. The RGB pixel values were then converted into the Commission International de l'Eclairage (CIE) LAB color space, and the color information was subsequently discarded. The luminance (L*) pattern was passed through a bank of three-dimensional spatiotemporal Gabor wavelet filters. The outputs of the two filters with orthogonal phases (quadrature pairs) were squared and summed to yield local ME. ME was compressed with a log-transformation and temporally downsampled to 0.5 Hz. Filters were tuned to six spatial frequencies (0, 1.5, 3.0, 6.0, 12.0, 24.0 cycles per image) and three temporal frequencies (0, 4.0, 8.0 Hz), without directional parameters. Filters were positioned on a square grid that covered the screen. The adjacent filters were separated by 3.5 standard deviations of their spatial Gaussian envelopes. The total number of ME features was 1,395.

**Modulation transfer function features**. A sound cochleogram was generated using a bank of 128 overlapping bandpass filters ranging from 20 to 10,000 Hz. The window size was set to 25 ms and the hop size to 10 ms. The filter output was averaged across 2 s (TR). We further extracted the features from the MTF model[62], which we provided in a public repository (https://osf.io/ea2jc/). For each cochleogram, a convolution with modulation-selective filters was then calculated. The outputs of the two filters with orthogonal phases (quadrature pairs) were squared and summed to yield the local modulation energy. Modulation energy was then log-transformed, averaged across 2 s, and further averaged within each of the 10 nonoverlapping frequency ranges logarithmically spaced along the frequency axis. The filter outputs of the upward and downward sweep directions were used. Modulation-selective filters were tuned to five spectral modulation scales (0.50, 1.0, 2.0, 4.0, 8.0 cycles per octave) and five temporal modulation rates (4.0, 8.0, 16.0, 32.0, 64.0 Hz). The total number of MTF features was 1,000.

**Button response features**. The BR features were constructed based on the number of button responses within 1 s for each of the four buttons, with the right two buttons pressed by the right thumb and the left two buttons pressed by the left thumb. The total number of BR features was four.

**Decoding model fitting**. In the decoding model, the cortical response matrix $\mathbf{R_D}$ [T × 3 V] was modeled using concatenating sets of [T × V] matrices with temporal delays of 2, 4, and 6 s. The feature matrix $\mathbf{F_D}$ [T × N] was modeled by multiplying the cortical response matrix $\mathbf{R_D}$ with the weight matrix $\mathbf{W_D}$ [3 V × N]:

$$\hat{\mathbf{F}}_\mathbf{D} = \mathbf{R_D}\mathbf{W_D} \qquad (2)$$

The weight matrix $\mathbf{W_D}$ was estimated using an L2-regularized linear regression with the training dataset, following the same procedure for the encoding model fitting.

**Decoding with novel tasks**. In order to examine the generalizability of our models, we performed encoding and decoding analyses with novel tasks not used

during model training (Fig. 5a). We randomly divided the 103 tasks into five task groups. A single task group contained 20–21 tasks. We performed five independent model fittings, each with a different task group as the target. From the training dataset, we excluded the time points during which the target tasks were performed and those within 6 s after the presentation of the target tasks. In the test dataset, we used only the time points during which the target tasks were performed and those within 6 s after the presentation of the target tasks. This setting allowed us to assume that the activity induced by the target task group and that induced by the other four task groups (training task groups) did not overlap, thus enabling us to investigate prediction and decoding accuracies for novel tasks. We performed encoding and decoding model fitting with the training task group, which was composed of 82–83 tasks. For model testing, we concatenated the predicted responses or decoded features of the five task groups. Responses or features for the time points that were duplicated were then averaged across the five task groups. Note that encoding and decoding with the novel tasks were only possible with the cognitive factor model because the original tasks needed to be transformed into the latent feature space.

For the decoding analysis with novel tasks, we measured the similarity between the CTF of each task and each decoded cognitive factor vector using Pearson's correlation coefficients for each time point. We refer to the correlation coefficient as the *task score*. We then calculated the time-averaged task scores for each task, and then performed decoding using the one-vs.-one method. For each target task, a series of binary classifications were performed between the target task and each of the remaining 102 tasks. The decoding accuracy was then calculated as a ratio that the target task had higher task score in this procedure. The one-vs-one decoding accuracy (ranged [0, 1], chance level = 0.5) can be considered as a standardized measure of the task score-based decoding accuracy (ranged [−1, 1], chance level = 0).

**Reconstruction of cortical activity from cerebellar and subcortical activities**. To evaluate whether the cerebellum and subcortex contain rich information to reconstruct cortical activity, we constructed additional encoding models using the cerebellar and subcortical activities as feature matrices. We first applied PCA to the original brain response matrices of the cerebellum and subcortex to reduce their dimensions to 2,000. Response matrices of the cerebellum and subcortex were independently used in the cerebellum and subcortex encoding models, respectively. For the cerebellum + subcortex encoding model, concatenated brain response matrix (totally 4,000 dimensions) was used as a feature matrix. The above encoding models were constructed and evaluated similarly as described in the encoding model fitting section.

**Statistics and reproducibility**. Statistical significance of encoding models (412 samples, one-sided) was computed by comparing estimated correlations with the null distribution of correlations between the two independent Gaussian random vectors with the same length as the test dataset[38]. The statistical threshold was set at $P < 0.05$ and corrected for multiple comparisons using the FDR procedure[63]. The statistical significance of the decoding accuracy (for both one-vs-one and task score-based methods) was tested for each task using the one-sided sign test (102 samples; $P < 0.05$, with FDR correction)[64], which is a nonparametric statistical test used in previous decoding studies[45,65–67]. The significance of average decoding accuracy (across tasks) was further tested using the permutation test ($P < 0.05$). Specifically, task labels in the test dataset were randomly shuffled 5,000 times, and p-values were calculated (for both one-vs-one and task score-based methods) based on the resultant null distribution of average decoding accuracy. Results from the six subjects were considered as replications of the analyses. All model fitting and analyses were conducted using custom software written on MATLAB. For data visualization on the cortical maps, pycortex was used[68].

**Reporting summary**. Further information on research design is available in the Nature Portfolio Reporting Summary linked to this article.

## Data availability

The raw MRI data are available at the OpenNeuro.org (https://openneuro.org/datasets/ds002306)[69]. Source data underlying Figs. 2d–f, 3b, 4b, 5b–e, and 6f are provided in Supplementary Data 1, 2, 3, 4, and 5, respectively. Other data are available from the corresponding author upon reasonable request.

## Code availability

The MATLAB code for building encoding and decoding models are available at the Open Science Framework (OSF) repository associated with our previous study[45] (https://osf.io/ea2jc/). Additional codes used in the current study and the datasets generated and/or analyzed during the current study are available upon request.

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

## Acknowledgements
We thank MEXT/JSPS KAKENHI (grant numbers JP17K13083 and JP18H05091 in #4903 (Evolinguistics) for T.N., JP15H05311 and JP18H05522 for S.N.) as well as JST CREST JPMJCR18A5 and ERATO JPMJER1801 (for S.N.) for the partial financial support to this study. The funders had no role in the study design, data collection and analysis, decision to publish, or preparation of the manuscript.

## Author contributions
T.N. and S.N. designed the study; T.N. collected and analyzed the data; T.N. and S.N. wrote the manuscript.

## Competing interests
The authors declare no competing interests.
