## [Peer Review File · Communications Biology]

Reviewers' comments:

Reviewer #1 (Remarks to the Author):

This is a very interesting study from T. Nakai & S. Nishimoto on the differential representation of several cognitive functions across the cortex, cerebellum and subcortex.

This study comes to complete the previous one of the authors published in 2020 (Nat Commun). In the present study, the authors go beyond their previous results in order to assess if the cerebellum and the subcortex have sufficient information to reconstruct cerebral cortex activity. The manuscript is well written and difficult to discuss since already validate (i.e., the method used in this manuscript is really close to the one used in the previous publication). Therefore, I recommend the publication of the manuscript after correcting several minor points:

1- The last section of the introduction really looks like an abstract and does not really address the relevance of the present work.

2- As for FIG-2D-2E, could the authors provide p-values in the FIG-5C-5D.

3- Page 13 line 223, the authors wrote "decoding accuracy, mean \pm SD, 95.2% \pm 0.9%; 99.5% \pm 0.5% ..." I am not sure to really understand what are these values. Does the first one correspond to the lower confidence interval and the second to the upper confidence interval? Could the authors indicate more clearly the meaning of these values?

4- Page 20 line 386, the authors referred to supplementary methods, but no additional methods are given in the supplementary file.

Reviewer #2 (Remarks to the Author):

This is an interesting manuscript with potential relevance to many subfields of neuroscience. The authors have provided a large-scale analysis of the responses of many brain regions to a range of cognitive tasks, using encoding models to assess how the structure of tasks is represented within cortex, subcortex, and cerebellum. They nicely demonstrate the similarity of task representations across these areas, as well as showing that novel tasks can be decoded from each area using a continuous latent task feature space.

I have just a few concerns that the authors should try to address:

1. The final analysis, where the authors try to predict cortical voxel activation from voxels in cerebellum and subcortex (voxel-to-voxel model) is potentially problematic due to the fact that voxels across the brain likely have strongly correlated noise. For example, since the overall signal in the brain fluctuates across trials, the trial-to-trial fluctuations on two voxels anywhere in the brain are likely to be very similar, even if these voxels have very little similarity in their stimulus/task encoding properties. I suspect that such noise correlations is part of the reason why the voxel-to-voxel encoding model works so well. One way to address this would be to subtract out the average signal (across the entire brain) from all voxels at each timepoint, as this should reduce the correlations. Another approach might be to shuffle the data from trials in a given task condition (i.e. maintaining task groupings but randomizing the order of trials within a task), as this should also break up the possibility of correlated noise. Some demonstration that the model performance is not just due to correlated noise will be needed to make this analysis convincing.

2. The control for sensorimotor voxels needs a bit more justification:

a. For example, why are these feature spaces chosen, and why are they thought to be sufficient? Why is a relatively simple visual model (motion energy) chosen over more expressive feature spaces such as a deep neural network model (i.e. ResNet)?

b. It would also be helpful to see some quantification of how many voxels are removed when the sensorimotor voxels are identified, and in which areas. For example, one might expect that this

control should remove most of the voxels in primary visual and primary auditory cortex, leaving mainly voxels in higher association cortex areas. Is such a pattern evident? If not, and if a lot of early sensory voxels are still retained in this analysis, it calls into question whether this analysis has fully controlled for sensory-evoked responses.

c. Finally, many voxels may encode information related to both sensory inputs and task-related information, especially in association cortex areas. The identification of voxels as being either "sensorimotor" or not makes an implicit assumption that voxels either code for sensory inputs or task factors, but not both. This issue should be acknowledged.

3. The principal components analysis figures (Figures 3 and 4) are a bit difficult to visually understand. For example, the white text labels are overlapping in the first panels which makes them hard to read, and the color scheme is a bit hard to interpret. Perhaps the labels can be made larger or a select few labels can be chosen. The interpretation of the first three components in the legend "[PC1= auditory (red); PC2 =audiovisual (green); PC3 = language 189 (blue)]" is also a bit confusing. Where do these interpretations come from, are they supposed to be evident from looking at the figure? Some more time should be spent unpacking and justifying these interpretations of the axes in the text. It could also be helpful to directly place those verbal labels on the figure somewhere, to indicate which "clusters" of tasks are meant to fall into which groups (maybe put a circle around groups of similar tasks?). This would also help with the interpretation of panels C and D.

4. Some more details are needed for several methodological points:

a. For the decoding analyses – it's not entirely clear from the methods what the difference is between the one-vs-one accuracy (lines 222-231) and the task score accuracy (lines 232 - 241). If the task score is a correlation coefficient, how do you get to an accuracy value? And why are the accuracy values so high for that method, as compared to the other method? From the histograms, it looks like the task scores ought to be lower than the one-vs-one decoding.

b. More detail/background on the Neurosynth database method is needed – what is the "reverse inference" map and what does it show?

c. On line 514, the phrase "for each of the 715 terms selected" is not clear - what are the 715 terms that were selected?

d. On lines 502-509, there is a description of a method for making a PCA score map for each subject – however, I don't think this figure is shown anywhere (maybe I missed it).

We greatly appreciate the reviewers' time and effort in reviewing our manuscript and providing helpful comments. Based on their comments, we have revised the original manuscript. Please find our responses to the comments below, along with the related changes in the revised manuscript.

Reviewer #1 (Remarks to the Author):

Comment 1-1: 1- The last section of the introduction really looks like an abstract and does not really address the relevance of the present work.

Response 1-1: Based on the reviewer's suggestion, we simplified the last paragraph of the Introduction, as follows, and merged it with the previous paragraph:

“To address these issues, we reanalyzed our previous fMRI data (Nakai and Nishimoto, 2020) and constructed voxel-wise encoding and decoding models independently using cortical, cerebellar, and subcortical voxels (Fig. 1a). The current approach reveals representations of abstract cognitive functions not only in the cerebral cortex but also in the cerebellum and subcortex.”

We also changed Figs. 1c and 1d to Figs. 5a and 6a, respectively. We modified the first sentence in the **Decoding of novel cognitive tasks from the activity of the cortex, cerebellum, and subcortex** subsection as well:

“To examine the specificity of how multiple cognitive tasks were represented in the different parts of the brain, we constructed decoding models for each, the cerebral cortex, cerebellum, and subcortex (Fig. 5a).”

In the **Reconstruction of cortical activity from cerebellar and subcortical activities** subsection:

“Finally, we tested whether the activity in the cerebral cortex can be reconstructed from activities in the cerebellum and subcortex (Fig. 6a).”

Comment 1-2: 2- As for FIG-2D-2E, could the authors provide p-values in the FIG-5C-5D.

Response 1-2: We added p values in Figs. 5c, 5d, and in Supplementary Figs. 9, 10.

Comment 1-3: 3- Page 13 line 223, the authors wrote "decoding accuracy, mean \pm SD, 95.2% \pm 0.9%; 99.5% \pm 0.5% ..." I am not sure to really understand what are these values. Does the first one correspond to the lower confidence interval and the second to the upper confidence interval? Could the authors indicate more clearly the meaning of these values?

Response 1-3: In our method, decoding accuracy was calculated for each of the 103 tasks. Therefore, we obtained a distribution of the decoding accuracy (103 values, as shown in the histogram Fig. 5b). The first value corresponds to the mean \pm SD of these decoding accuracy values.

For each task, we further tested whether the obtained decoding accuracy was significant or not (one-sided sign test, $P < 0.05$, FDR-corrected). Thus, we obtained a percentage of significantly decoded tasks. This is the second value.

To show more clearly the difference between these two values, we separated the first descriptions of the two different values in the **Decoding of novel cognitive tasks from the activity of the cortex, cerebellum, and subcortex** subsection as follows:

"The decoding model of the cerebral cortex significantly decoded more than 95% of the cognitive tasks (decoding accuracy, mean \pm SD, 95.2% \pm 0.9%; Fig. 5b top) (see Supplementary Fig. 7 for the data of the other subjects). Significance of the decoded accuracy was further evaluated using a one-sided sign test, and more than 99% of tasks were significantly decoded (99.5% \pm 0.5% of the tasks were significant; $P < 0.05$, false discovery rate [FDR]-corrected)."

Comment 1-4: 4- Page 20 line 386, the authors referred to supplementary methods, but no additional methods are given in the supplementary file.

Response 1-4: Thank you for the indication. We added this information of 103 tasks in the Supplementary Information of the new manuscript.

Reviewer #2 (Remarks to the Author):

Comment 2-1: 1. The final analysis, where the authors try to predict cortical voxel activation from voxels in cerebellum and subcortex (voxel-to-voxel model) is potentially problematic due to the fact that voxels across the brain likely have strongly correlated noise. For example, since the overall signal in the brain fluctuates across trials, the trial-to-trial fluctuations on two voxels anywhere in the brain are likely to be very similar, even if these voxels have very little similarity in their stimulus/task encoding properties. I suspect that such noise correlations is part of the reason why the voxel-to-voxel encoding model works so well. One way to address this would be to subtract out the average signal (across the entire brain) from all voxels at each timepoint, as this should reduce the correlations. Another approach might be to shuffle the data from trials in a given task condition (i.e. maintaining task groupings but randomizing the order of trials within a task), as this should also break up the possibility of correlated noise. Some demonstration that the model performance is not just due to correlated noise will be needed to make this analysis convincing.

*This comment is also related to the Editor's comment: while we ask to take into account all the reviewers' suggestions, we wanted to stress that controlling for the average signal when assessing the voxel-to-voxel correlation.

Response 2-1: Thank you for the important indication. We agree with the reviewer's (and editor's) point. We thus performed an additional voxel-to-voxel encoding model analysis after subtracting mean signal across the entire brain in each time point from the original response of the cortex, cerebellum, and subcortex. We found similar prediction performance in this analysis (prediction accuracy of the cerebellum model, 0.365 ± 0.020 ; the subcortex model, 0.295 ± 0.016 ; the cerebellum + subcortex model, 0.365 ± 0.029). Although we observe some decrease in the prediction performance, we think that our results are not entirely explained by the correlated noise.

Regarding this analysis, we added the following text in the **Reconstruction of cortical activity from cerebellar and subcortical activities** subsection:

“To exclude the possibility that the prediction performances of cerebellum and subcortex models are caused by correlated noise among three brain parts, we constructed additional encoding models after subtracting average brain response across the entire brain in each time point. These models again predicted activity in most cortical regions (prediction accuracy of the cerebellum model, 0.365 ± 0.020 ; the subcortex model, 0.295 ± 0.016 ; the cerebellum + subcortex model,

0.365 ± 0.029).”

Comment 2-2: 2. The control for sensorimotor voxels needs a bit more justification:

a. For example, why are these feature spaces chosen, and why are they thought to be sufficient? Why is a relatively simple visual model (motion energy) chosen over more expressive feature spaces such as a deep neural network model (i.e. ResNet)?

Response 2-2: We chose the motion energy (visual) and modulation transfer function (auditory) models because we also used these models in our previous papers and already confirmed their efficacy (Nakai and Nishimoto, Nat Commun 2020; Koide-Majima et al., NeuroImage 2021; Nakai et al, Cerebral Cortex 2021). Moreover, these models are easier to interpret because they were explicitly modeled to mimic neuronal response patterns in the early visual and auditory cortices. We clarified these points in the **Exclusion of sensorimotor voxels** subsection in the new manuscript:

“Our previous studies validated the efficacy of these feature in controlling sensorimotor information^{33,44,45}. Moreover, ME and MTF features are easier to interpret than neural network features because they explicitly modeled neuronal response patterns in the early visual and auditory cortices.”

Comment 2-3: b. It would also be helpful to see some quantification of how many voxels are removed when the sensorimotor voxels are identified, and in which areas. For example, one might expect that this control should remove most of the voxels in primary visual and primary auditory cortex, leaving mainly voxels in higher association cortex areas. Is such a pattern evident? If not, and if a lot of early sensory voxels are still retained in this analysis, it calls into question whether this analysis has fully controlled for sensory-evoked responses.

Response 2-3: By quantifying the number of removed voxels, we found large variability in the number of removed voxels among the cortex, cerebellum, and subcortex. With threshold = 0.3, 18.2% ± 5.0% of cortical voxels were excluded. However, only 1.8% ± 0.8% and 0.1% ± 0.1% of voxels were excluded in the cerebellum and subcortex, respectively. The current threshold (0.3) of sensorimotor voxels might be too high for the cerebellum and subcortex.

Thus, we quantified sensorimotor voxels with different thresholds. With threshold = 0.2,

35.3% \pm 9.1%, 7.0% \pm 2.7%, and 0.5% \pm 0.2% of cortical, cerebellar, and subcortical voxels were respectively excluded. With threshold = 0.1, 63.4% \pm 10.1%, 33.3% \pm 9.1%, and 8.8% \pm 3.6% of cortical, cerebellar, and subcortical voxels were respectively excluded.

For all the thresholds tested, we found a significant correlation between task similarities of the cortex and cerebellum (threshold = 0.2, $\rho = 0.820$; threshold = 0.1, $\rho = 0.785$), as well as the cortex and subcortex (threshold = 0.2, $\rho = 0.669$; threshold = 0.1, $\rho = 0.662$).

For all the thresholds tested, we found that most tasks were significantly decoded using the cortical voxels (threshold = 0.2; mean \pm SD, 94.6% \pm 0.9%; 98.9% \pm 0.7% of the tasks were significant; threshold = 0.1; mean \pm SD, 93.1% \pm 1.2%; 97.6% \pm 1.2% of the tasks were significant), cerebellar voxels (threshold = 0.2; mean \pm SD, 90.9% \pm 1.5%; 96.0% \pm 3.0% of the tasks were significant; threshold = 0.1; mean \pm SD, 88.6% \pm 1.4%; 93.5% \pm 3.0% of the tasks were significant), and subcortical voxels (threshold = 0.2; mean \pm SD, 85.4% \pm 1.2%; 91.9% \pm 1.6% of the tasks were significant; threshold = 0.1; mean \pm SD, 84.6% \pm 1.4%; 90.0% \pm 1.7% of the tasks were significant).

For all thresholds tested, positive correlations of decoding performances were found after excluding the sensorimotor voxels (threshold = 0.2; between the cortex and cerebellum, $\rho = 0.839 \pm 0.032$; cortex and subcortex, $\rho = 0.695 \pm 0.089$; threshold = 0.1; between the cortex and cerebellum, $\rho = 0.825 \pm 0.051$; cortex and subcortex, $\rho = 0.711 \pm 0.093$).

These analyses indicate that the current results are not much affected by the thresholds that had defined sensorimotor voxels.

We described the task similarities and decoding results with different thresholds in Supplementary Information:

“Task similarities after excluding sensorimotor voxels with other thresholds

In all thresholds, we found a significant correlation between task similarities of the cortex and cerebellum (threshold = 0.2, $\rho = 0.820$; threshold = 0.1, $\rho = 0.785$), as well as the cortex and subcortex (threshold = 0.2, $\rho = 0.669$; threshold = 0.1, $\rho = 0.662$).

Decoding after excluding sensorimotor voxels with other thresholds

With threshold = 0.2, most tasks were significantly decoded using the cortical voxels (threshold = 0.2; mean \pm SD, 94.6% \pm 0.9%; 98.9% \pm 0.7% of the tasks were significant; threshold = 0.1; mean \pm SD, 93.1% \pm 1.2%; 97.6% \pm 1.2% of the tasks were significant), cerebellar voxels (threshold = 0.2; mean \pm SD, 90.9% \pm 1.5%; 96.0% \pm 3.0% of the tasks were significant; threshold = 0.1; mean \pm SD, 88.6% \pm 1.4%; 93.5% \pm 3.0% of the tasks were significant), and subcortical

voxels (threshold = 0.2; mean \pm SD, 85.4% \pm 1.2%; 91.9% \pm 1.6% of the tasks were significant; threshold = 0.1; mean \pm SD, 84.6% \pm 1.4%; 90.0% \pm 1.7% of the tasks were significant). Positive correlations of decoding performances were again found between the cortex and cerebellum ($\rho = 0.839 \pm 0.032$) and between the cortex and subcortex ($\rho = 0.695 \pm 0.089$).

With threshold = 0.1, most tasks were significantly decoded using the cortical voxels (mean \pm SD, 93.1% \pm 1.2%; 97.6% \pm 1.2% of the tasks were significant), cerebellar voxels (mean \pm SD, 88.6% \pm 1.4%; 93.5% \pm 3.0% of the tasks were significant), and subcortical voxels (threshold = 0.1; mean \pm SD, 84.6% \pm 1.4%; 90.0% \pm 1.7% of the tasks were significant). Positive correlations of decoding performances were again found between the cortex and cerebellum ($\rho = 0.825 \pm 0.051$) and between the cortex and subcortex ($\rho = 0.711 \pm 0.093$).”

In the Brain representations of task structures were preserved across the cortex, cerebellum, and subcortex subsection in the main manuscript:

“We found similar results with the other thresholds ($r = 0.2, 0.1$) for selecting sensorimotor voxels (Supplementary Information).”

Next, we quantified the removed voxels in several anatomical regions chosen from association and early sensory cortices. We selected the inferior frontal gyrus (IFG) and inferior parietal lobule (IPL) as representatives of association cortices, as well as Heschl’s gyrus (HG) and occipital pole (OP) as representatives of early sensory cortices.

We found that a relatively larger ratio of voxels was excluded from the association cortices (for the threshold of 0.3; 8.2% \pm 6.1% from the left IFG, 23.7% \pm 11.3% from the left IPL) compared to early sensory cortices (e.g., 54.7% \pm 17.2% from the left HG, 57.7% \pm 5.9% from the left OP). This was also clearly found in the cortical map of removed voxels (newly added Supplementary Figure 1).

With threshold = 0.2, 26.6% \pm 16.5% from the left IFG, 52.3% \pm 18.0% from the left IPL, 74.0% \pm 12.6% from the left HG, 73.0% \pm 6.2% from the left OP were removed. With threshold = 0.1, 67.0% \pm 18.2% from the left IFG, 84.5% \pm 9.0% from the left IPL, 91.3% \pm 6.7% from the left HG, 86.3% \pm 7.6% from the left OP were removed.

These analyses indicate that the control analysis predominantly excluded early sensory regions than higher-order association cortices.

To summarize the ratios of removed voxels, we added Supplementary Table 1 in the Supplementary Information.

“Supplementary Table 1. Ratios of the removed sensorimotor voxels in representative regions

Regions	Thresholds		
	0.3	0.2	0.1
Whole cortex	18.2% ± 5.0%	35.3% ± 9.1%	63.4% ± 10.1%
Whole cerebellum	1.8% ± 0.8%	7.0% ± 2.7%	33.3% ± 9.1%
Whole subcortex	0.1% ± 0.1%	0.5% ± 0.2%	8.8% ± 3.6%
Left IFG	8.2% ± 6.1%	26.6% ± 16.5%	67.0% ± 18.2%
Right IFG	13.0% + 10.2%	34.8% + 17.4%	73.4% + 13.9%
Left IPL	23.7% ± 11.3%	52.3% ± 18.0%	84.5% ± 9.0%
Right IPL	25.7% + 10.7%	54.7% + 13.2%	86.0% + 5.6%
Left HG	54.7% ± 17.2%	74.0% ± 12.6%	91.3% ± 6.7%
Right HG	56.7% + 17.7%	75.7% + 12.8%	93.9% + 6.1%
Left OP	57.7% ± 5.9%	73.0% ± 6.2%	86.3% ± 7.6%
Right OP	59.3% + 9.1%	72.8% + 8.0%	86.5% + 4.8%

IFG, inferior frontal gyrus; IPL, inferior parietal lobule; HG, Heschl’s gyrus; OP, occipital pole.”

We added Supplementary Figure 1 and its legend to show the removed voxels (in the cerebral cortex) in this analysis:

“Supplementary Figure 1. Cortical map of sensorimotor voxels predicted by the regressor features (threshold, $r = 0.3$), mapped on the flattened cortical sheets of subjects ID01–ID06.”

Comment 2-4: c. Finally, many voxels may encode information related to both sensory inputs and task-related information, especially in association cortex areas. The identification of voxels as being either “sensorimotor” or not makes an implicit assumption that voxels either code for sensory inputs or task factors, but not both. This issue should be acknowledged.

Response 2-4: We agree with the reviewer regarding the concern of voxels related to sensorimotor and task information. However, we did not have such implicit assumption and have discussed the possible effect of voxels that may code sensorimotor and task-related information in the Discussion section of the original manuscript:

“To exclude the possibility that similar task representations were caused simply by low-level sensorimotor components, we performed additional analysis by excluding voxels that were predictable by the sensorimotor features. This analysis was more conservative than the analysis

adopted in our previous study, which concatenated sensorimotor features with the target features (Nakai and Nishimoto, 2020) or quantitatively assessed the ratio of explained variances (de Heer et al., 2017; Nakai et al., 2021b). This is because the current approach excludes voxels that may contain partial sensorimotor information. In other words, the current approach more robustly ruled out the possible contribution of sensorimotor features to the representational similarity and decodability across the three brain parts.”

The current method, which excludes voxels that may only partially represent sensorimotor information, might be too conservative and underestimate the representational similarity and decodability across the three brain parts. Conversely, the fact that the similar task representations and decodability were obtained even under such conservative conditions suggests that the results of this study are robust.

We explained this point more clearly in the **Discussion** section in the new manuscript:

“This analysis was more conservative than the analysis adopted in our previous study, which concatenated sensorimotor features with the target features⁴⁵ or quantitatively assessed the ratio of explained variances^{33,39}. This is because the current approach excludes voxels which may contain both sensorimotor and task-related information. Although the current approach may underestimate representational similarity and decodability across the three brain parts, it more robustly ruled out the possible contribution of sensorimotor features.”

Comment 2-5: 3. The principal components analysis figures (Figures 3 and 4) are a bit difficult to visually understand. For example, the white text labels are overlapping in the first panels which makes them hard to read, and the color scheme is a bit hard to interpret. Perhaps the labels can be made larger or a select few labels can be chosen.

Response 2-5: We agree with the reviewer that the PCA 2D maps were not intelligible enough. Thus, we reduced the task labels and left only 30 task labels. We also added the following text in the legends of Figures 3 and 4:

For a better visibility, only 30 tasks are shown in white.

Meanwhile, we think that we need to present all task labels in these 2D maps. Thus, we added Supplementary Figures 3 and 4 in the Supplementary Information, showing the same figures as Figures 3a and 4a, but with all task labels.

We added the following text in **Visualization of cognitive structures in each anatomical and functional subregion** subsection:

“(Cerebellum, Fig. 3a, Supplementary Figure 3; Subcortex, Fig. 4a, Supplementary Figure 4).”

We added the following text in the Supplementary Information:

“Supplementary Figure 3. Visualization of task structures in the cerebellum. All tasks are labeled.”

“Supplementary Figure 4. Visualization of task structures in the subcortex. All tasks are labeled.”

Comment 2-6: The interpretation of the first three components in the legend “[PC1= auditory (red); PC2 =audiovisual (green); PC3 = language 189 (blue)]” is also a bit confusing. Where do these interpretations come from, are they supposed to be evident from looking at the figure? Some more time should be spent unpacking and justifying these interpretations of the axes in the text.

Response 2-6: Thank you for pointing out our mistake. These interpretations are the PCs of the cerebral cortex (which we had shown in Nakai and Nishimoto, Nat Commun 2020) but not those of the cerebellum and subcortex. Therefore, we removed these interpretations from the legends in Figures 3 and 4:

“Colors indicate the loadings of the top three principal components [PC1, red; PC2, green; PC3, blue]”

However, we think that we should provide interpretations for the principal components of the cerebellum and subcortex. We thus performed a metadata-based reverse inference analysis of principal components (Nakai and Nishimoto, Nat Commun 2020) in the **Principal component analysis of task-type weights** subsection:

“To obtain an objective interpretation of the PCs, we performed metadata-based inference of the cognitive factors related to each PC. We used Neurosynth as a metadata reference of the past neuroimaging literature⁴⁶. From the approximately 3,000 terms in the database, we manually selected 715 terms that covered the comprehensive cognitive factors while also avoiding redundancy. In particular, we removed several plural terms that also had their singular counterpart (e.g., “concept” and “concepts”) and past tense verbs that also had their present counterpart (e.g., “judge” and “judged”) from the dataset. We also excluded those terms that indicated anatomical regions (e.g., “parietal”). We used the reverse inference map of the Neurosynth database for each of the 715 selected terms. The reverse inference map indicated the

likelihood of a given term being used in a study if the activity was observed at a particular voxel. Each reverse inference map in the MNI152 space was then registered to the subjects' reference echo planar imaging (EPI) data using FreeSurfer^{53,54}. For each of the PCA score maps, we calculated Pearson's correlation coefficients with the 715 registered reverse inference maps, which resulted in a cognitive factor vector with 715 elements. Terms with higher correlation coefficient values were regarded as contributing more to the target PC."

We reported the following interpretation results of cerebellar and subcortical PCs in the main manuscript and added Supplementary Tables 2, 3:

"Metadata-based interpretation of principal components in the cerebellum and subcortex To interpret cognitive factors related to PCs, we performed a metadata-based reverse inference analysis. For each of the PC score maps, we calculated Pearson's correlation coefficients with the 715 reverse inference maps taken from the Neurosynth database⁴⁶. The top and bottom 10 terms of each PC provided their objective interpretations (Supplementary Tables 2, 3). For the sake of intelligibility, we only presented the results of the top five PCs.

For the cerebellum, PC1 was associated with introspection and emotion terms on the positive side (top 10 terms; "theory mind," "disgust") and executive function and motor terms on the negative side (bottom 10 terms; "working memory," "motor"). Contrarily, PC2 was associated with executive function terms on the positive side ("working memory," "execution") and introspection terms on the negative side ("autobiographical," "theory mind"). PC3 was associated with language terms on the positive side ("sentence," "comprehension") and motor terms on the negative side ("motor," "finger"). PC4 was associated with motor terms on the positive side ("finger," "sensorimotor") and language terms on the negative side ("reading," "linguistic"). PC5 was associated with cognitive demand terms on the positive side ("cognitive task," "calculation") and motor terms on the negative side ("finger," "motor").

For the subcortex, PC1 was associated with emotion terms on the positive side ("emotion," "valence"), whereas it was associated with motor terms on the negative side ("finger," "movement"). PC2 was associated with memory terms on the positive side ("memory," "retrieval") and somatosensory terms on the negative side ("pain," "somatosensory"). PC3 was associated with motor terms on the positive side ("muscle," "finger") and emotion terms on the negative side ("pain," "emotion"). PC4 was also associated with motor terms on the positive side ("preparation," "motor") and somatosensory terms on the negative side ("pain," "somatosensory"). PC5 was associated with memory terms on the positive side ("retrieval," "memory") and motor terms on the negative side ("finger," "sensorimotor").

These results indicate that although abstract cognitive functions are represented in the cerebellum and subcortex, their distributions are distinct. Introspection, executive function, motor function, and language are predominantly represented in the cerebellum, whereas motor function, emotion, and memory are predominantly represented in the subcortex.

Supplementary Table 2. Top cognitive factors related to each principal component of the cerebellum

		Top and bottom cognitive factors in the Neurosynth database
PC1	Top	“self referential,” “belief,” “word form,” “disgust,” “visual word,” “mind,” “self,” “theory mind,” “face,” “anger”
	Bottom	“execution,” “working memory,” “verbal working,” “motor,” “imagery,” “memory,” “rehearsal,” “motor imagery,” “visual motion,” “muscle”
PC2	Top	“rehearsal,” “face,” “phonological,” “verbal,” “working memory,” “visual,” “verbal working,” “execution,” “reading,” “task”
	Bottom	“autobiographical,” “autobiographical memory,” “episodic,” “mentalizing,” “mind tom,” “self referential,” “theory mind,” “aging,” “nervous,” “tom”
PC3	Top	“autobiographical,” “sentence,” “content,” “semantic,” “mind,” “reading,” “comprehension,” “empathy,” “default,” “theory mind”
	Bottom	“motor,” “finger,” “movement,” “sensorimotor,” “hand,” “finger movements,” “index finger,” “finger tapping,” “tapping,” “muscle”
PC4	Top	“finger,” “object,” “index finger,” “sensorimotor,” “motor,” “finger movements,” “hand,” “somatosensory,” “movement,” “finger tapping”
	Bottom	“rehearsal,” “reading,” “working memory,” “verbal working,” “verbal,” “sentence,” “linguistic,” “wm task,” “phonological,” “memory”
PC5	Top	“cognitive task,” “phonological,” “semantic,” “calculation,” “impulsivity,” “verbal,” “face,” “navigation,” “executive,” “modality”
	Bottom	“finger,” “motor,” “movement,” “sensorimotor,” “hand,” “tapping,” “finger tapping,” “somatosensory,” “index finger,” “execution”

The top and bottom 10 cognitive factors in the Neurosynth database for PC1–PC5, based on the correlation coefficients between each PC score map and the 715 registered reverse inference maps.

Supplementary Table 3. Top cognitive factors related to each principal component of the subcortex

		Top cognitive factors in the Neurosynth database
PC1	Top	“face,” “encoding,” “emotional,” “happy,” “neutral,” “emotion,” “valence,” “angry,” “episodic memory,” “episodic”
	Bottom	“finger,” “finger tapping,” “tapping,” “sensorimotor,” “execution,” “finger movements,” “motor imagery,” “monetary,” “pain,” “movement”
PC2	Top	“encoding,” “episodic,” “memory,” “autobiographical,” “episodic memory,” “retrieval,” “alzheimer,” “alzheimer disease,” “autobiographical memory,” “subsequent memory”
	Bottom	“pain,” “somatosensory,” “secondary somatosensory,” “noxious,” “painful,” “chronic pain,” “heat,” “finger tapping,” “finger,” “tapping”
PC3	Top	“muscle,” “nervous,” “sensorimotor,” “finger,” “motor performance,” “motor imagery,” “movement,” “motor task,” “pd,” “foot”
	Bottom	“pain,” “emotional,” “painful,” “task,” “face,” “aversive,” “threat,” “affective,” “emotion,” “happy”
PC4	Top	“preparation,” “vocal,” “preparatory,” “pseudowords,” “motor imagery,” “execution,” “parkinson,” “motor,” “finger movements,” “monetary”
	Bottom	“pain,” “somatosensory,” “secondary somatosensory,” “painful,” “positive negative,” “sexual,” “response inhibition,” “intensity,” “positive,” “heat”
PC5	Top	“retrieval,” “episodic,” “memory,” “episodic memory,” “encoding,” “autobiographical,” “retrieved,” “autobiographical memory,” “alzheimer,” “navigation”
	Bottom	“finger,” “movement,” “sensorimotor,” “motor,” “finger tapping,” “execution,” “finger movements,” “tapping,” “hand,” “motor imagery”

The top and bottom 10 cognitive factors in the Neurosynth database for PC1–PC5, based on the correlation coefficients between each PC score map and the 715 registered reverse inference maps.”

Comment 2-7: It could also be helpful to directly place those verbal labels on the figure somewhere, to indicate which “clusters” of tasks are meant to fall into which groups (maybe put a circle around groups of similar tasks?). This would also help with the interpretation of panels C and D.

Response 2-7: To facilitate the interpretation of Figures 3 and 4, we added labels of each principal component based on the metadata-based analysis described in the previous response. Thus, we

added “Introspection,” “Executive function,” and “Language” labels (corresponding to RGB colors) to Figure 3, and “Emotion,” “Memory,” and “Motor” labels to Figure 4.

In the legend of Figure 3, we added the following text:

“Each PC is labeled based on metadata-based interpretation analysis (Supplementary Table 1).”

In the legend of Figure 4, we added the following text:

“Each PC is labeled based on metadata-based interpretation analysis (Supplementary Table 2).”

Comment 2-8: 4. Some more details are needed for several methodological points:

a. For the decoding analyses – it’s not entirely clear from the methods what the difference is between the one-vs-one accuracy (lines 222-231) and the task score accuracy (lines 232 - 241). If the task score is a correlation coefficient, how do you get to an accuracy value? And why are the accuracy values so high for that method, as compared to the other method? From the histograms, it looks like the task scores ought to be lower than the one-vs-one decoding.

Response 2-8: The one-vs-one decoding accuracy is a standardized measure calculated based on the task score accuracy (Fig. 5a). We adopted this measure based on our previous research (Nakai and Nishimoto, Nat Commun 2020). The decoding models output correlation coefficient (i.e., task score) in each time bin. We then averaged the task score during each task, which resulted in 103 mean task scores.

One-vs-one accuracy is calculated for each task in the following way. Let’s assume that task-001 has a 0.6 task score. For each of the left-out 102 tasks, we compare its task score with that of the task-001: (1) task-002 (task score = 0.5) vs. task-001, (2) task-003 (task score = 0.8) vs. task-001, so on. If the task-001 wins against 90 out of 102 tasks, decoding accuracy = $90/102 = 0.88$.

The one-vs-one accuracy values were higher than task score accuracy values because the former values were calculated by standardizing the latter values (ranged $[-1, 1]$, chance level = 0) into the range of $[0, 1]$ with the chance level = 0.5. The chance level is higher in the former method.

We added the following text to explain details of the decoding accuracy method in the

Decoding with novel tasks subsection.

“The one-vs-one decoding accuracy (ranged $[0, 1]$, chance level = 0.5) can be considered as a standardized measure of the task score-based decoding accuracy (ranged $[-1, 1]$, chance level = 0).”

Comment 2-9: b. More detail/background on the Neurosynth database method is needed – what is the “reverse inference” map and what does it show?

Response 2-9: The reverse inference map indicates the likelihood of a given term (e.g., “phonological”) being used in a study (i.e., a paper included in the Neurosynth database) if the activity was observed at a particular voxel. To explain this, we added the following text in the **Principal component analysis of task-type weights** subsection:

“We used the reverse inference map of the Neurosynth database for each of the 715 selected terms. The reverse inference map indicated the likelihood of a given term being used in a study if the activity was observed at a particular voxel.”

Comment 2-10: c. On line 514, the phrase “for each of the 715 terms selected” is not clear - what are the 715 terms that were selected?

Response 2-10: We apologize for the lack of clarity. The 715 terms were the same as in our previous study (Nakai and Nishimoto, Nat Commun 2020). We manually selected those terms from ~3,000 original terms in the Neurosynth database (Yarkoni et al., Nat Methods 2011), because the original database has lots of duplications (such as “concept” and “concepts”) and noncognitive terms (“parietal”).

We described the selection criteria for Neurosynth terms in the **Principal component analysis of task-type weights** subsection:

“To obtain an objective interpretation of the PCs, we performed metadata-based inference of the cognitive factors related to each PC. We used Neurosynth as a metadata reference of the past neuroimaging literature⁴⁶. From the approximately 3,000 terms in the database, we manually selected 715 terms that covered the comprehensive cognitive factors while also avoiding redundancy. In particular, we removed several plural terms that also had their singular counterpart (e.g., “concept” and “concepts”) and past tense verbs that also had their present counterpart (e.g., “judge” and “judged”) from the dataset. We also excluded those terms that indicated anatomical regions (e.g., “parietal”).”

Comment 2-11: d. On lines 502-509, there is a description of a method for making a PCA score map for each subject – however, I don't think this figure is shown anywhere (maybe I missed it).

Response 2-11: Thank you for pointing out our mistake. This PCA score map was included in our previous paper (Nakai and Nishimoto, Nat Commun 2020; Figure 4a) but not in the current manuscript. Thus, we removed this paragraph.

In addition to the response to reviewers' comments, we also added the following text in **Methods** section indicated in the submission file checklist:

“Statistics and Reproducibility

Statistical significance of encoding models (412 samples, one-sided) was computed by comparing estimated correlations with the null distribution of correlations between the two independent Gaussian random vectors with the same length as the test dataset³⁸. The statistical threshold was set at $P < 0.05$ and corrected for multiple comparisons using the FDR procedure⁵⁷. The statistical significance of the decoding accuracy was tested for each task using the one-sided sign test (102 samples; $P < 0.05$, with FDR correction). Results from the six subjects were considered as replications of the analyses. All model fitting and analyses were conducted using custom software written on MATLAB. For data visualization on the cortical maps, pycortex was used⁵⁸.”

Reviewers' comments:

Reviewer #1 (Remarks to the Author):

The authors addressed all my comments.
Congratulations for this really interesting work.

Reviewer #2 (Remarks to the Author):

Overall, the authors have provided some nice revisions and improved the paper substantially. The analysis of subtracting the mean signal from all timepoints before performing the voxel-to-voxel encoding model is an appropriate way to address my concern from the previous version about noise correlations. The quantification and visualization of the sensorimotor voxels that were removed is also very thorough and addresses my concerns there. I still have a couple of concerns about the clarity of the manuscript and a few technical details that can hopefully be addressed without too much difficulty.

Main comments

The results section that describes the PCA analysis is still a bit hard to follow. In particular it is hard to tell from the text what is supposed to be the purpose of the new "metadata-based interpretation analysis" of the PCs, versus the visualization where each task is plotted in cognitive space. It seems like what you're trying to show is that you can recover an interpretation of what each PC represents based on using the NeuroSynth database, and that those interpretations independently line up with the clusters of tasks that you get by plotting the task organization in cognitive space (something along these lines?). It would be helpful to be more explicit about exactly what is the motivation for doing the metadata-based analysis, and if it agrees with the organization of the individual tasks in Fig 3a and 4a (I'm not convinced that it does, just from a glance). One suggestion here would be to move the section "Metadata-based interpretation of principal components in the cerebellum and subcortex" which now starts on line 186 to earlier in the results, for example around where line 138 is now. You could then describe everything that goes into Fig 3ab and Fig 4ab, including the PC interpretations, and after that go into describing the sub-region analyses that are in panels C-D as a separate paragraph.

Another comment about the PCA interpretation analysis – are there many voxels with a negative score for one or more of the PCs? If so, how should we interpret the cognitive factors that have negative weights ("bottom" terms in Supp Table 2)? It seems like these negatively weighted cognitive factors might have a meaningful interpretation, for example voxels with a negative score on PC1 should in theory be modulated by the cognitive factors that are negatively associated with PC1. Currently it seems like there is a focus only on the "positive" side of each PC, without addressing the "negative" side, so this warrants at least some discussion.

More discussion should be given to the differences between the cognitive task spaces encoded in the cerebellum versus subcortex and the cortex (Figs 3-4). For example, this sentence seems like an oversimplification: (line 212) "Introspection, executive function, motor function, and language are predominantly represented in the cerebellum, whereas motor function, emotion, and memory are predominantly represented in the subcortex." What about the cognitive factors that are negatively weighted for each PC? Also, if motor function is represented strongly by both cerebellum and subcortex, it doesn't make sense to say it is predominantly represented by either of them. Some text in the discussion should be devoted to the difference between these brain regions (and versus the cortex) and how it relates to past work.

Some methodological/clarifying points about the decoding results:

For the statistical testing on your decoding results (lines 225-245, and 644-646), some justification is needed on the choice of a one-sided sign test for evaluating significance (maybe there is a citation for this?). I am not sure this is an appropriately sensitive test for evaluating significance. A more traditional and robust method would be to use a permutation test, where you shuffle the labels of the dataset many times to generate a null distribution of accuracy, and then compare your real decoding accuracy to the null distribution to get a p value. You could do this for each of the tasks individually, or you could just compute a permuted p-value for the average decoding accuracy across all tasks (the latter might be simpler and less confusing for the reader).

Some clarity is also needed on how/whether decoding results were combined across subjects (line 646 "Results from the six subjects were considered as replications of the analyses."). Does this mean that you get a p-value for every individual subject? If so, are the "percent significant tasks" in the text (line 229) just for one subject? If they are, the values for every subject should be reported somewhere, or subject-averaged values should be reported.

I appreciate the authors' clarification on the point of the difference between the task score (which is a correlation coefficient) and the decoding accuracy. However I am still not following the analysis in lines 236-245. For example "we quantified decoding accuracy using task score". How are you quantifying decoding accuracy and computing significance from the task scores alone? It seems like in order to get decoding accuracy, you need to compare the task score against the other tasks, and that would be the same as the previous analysis (lines 225-235). Are you just comparing the task score versus its chance value (0.5)? If so, this doesn't seem like an appropriate method as this is not a robust way to evaluate significance (see notes above on permutation testing).

The interpretation of the difference in standard deviation between the correlation coefficients in the different brain areas needs more justification (Fig 2f, line 101) "...suggesting that the structure of cognitive tasks is more distinctively organized in the cortex compared to the cerebellum and subcortex".

The fact that the standard deviation is smaller for the cerebellum and subcortex than the cortex doesn't necessarily mean that there is more "distinctiveness" among the different tasks. It seems like the cortex has more large positive correlations between tasks than the other brain areas (for example in Fig 2d-e), which should mean that tasks are more similar there, not less distinctive? The difference in the distribution of correlation coefficients between weights across the different brain regions could also be due to differences in noise/signal quality across brain regions. These issues should be discussed.

Minor comments

Since the results text comes before methods here, it would be helpful to provide brief description of some of the major methods when they are first introduced in the text, or at least provide a parenthetical reference to the relevant section of the methods. For example, when you first mention the "sparse task type" model on line 88, you could have a brief note describing what the features of that model are so that the reader doesn't have to flip all the way to the methods section. Same can be done for the "latent cognitive factors" (line 221), and the "sensorimotor features" first mentioned at line 107.

In Figures 3 and 4, there should be a separate legend for panels C-D which indicate what the red/blue colors and dot sizes mean. Especially because the red/blue colors are also used in panel A, it looks as if the red/blue in the legend of A are referring to C-D. You could also consider changing to colors other than red/blue to help with the interpretability.

Line 161 - Supplementary Table 1 should be Supplementary Table 2

Line 622 - "We then calculated the time-averaged task scores for each task using the one-vs.-one method" - this is confusing, is this supposed to say something like "We then calculated the time-averaged task scores for each task, and then performed decoding using the one-vs.-one method." ?

Figure 6 - Some of the text is cut off in the titles of each panel, make sure to check this

Fig 6f - are error bars across subjects, or across voxels? Please clarify.

We greatly appreciate the reviewers' time and effort in reviewing our manuscript and providing helpful comments. Based on their comments, we have revised the original manuscript. Please find our responses to the comments below, along with the related changes in the revised manuscript.

Reviewer #2 (Remarks to the Author):

Comment 2-1: The results section that describes the PCA analysis is still a bit hard to follow. In particular it is hard to tell from the text what is supposed to be the purpose of the new “metadata-based interpretation analysis” of the PCs, versus the visualization where each task is plotted in in cognitive space. It seems like what you're trying to show is that you can recover an interpretation of what each PC represents based on using the NeuroSynth database, and that those interpretations independently line up with the clusters of tasks that you get by plotting the task organization in cognitive space (something along these lines?). It would be helpful to be more explicit about exactly what is the motivation for doing the metadata-based analysis, and if it agrees with the organization of the individual tasks in Fig 3a and 4a (I'm not convinced that it does, just from a glance). One suggestion here would be to move the section “Metadata-based interpretation of principal components in the cerebellum and subcortex” which now starts on line 186 to earlier in the results, for example around where line 138 is now. You could then describe everything that goes into Fig 3ab and Fig 4ab, including the PC interpretations, and after that go into describing the sub-region analyses that are in panels C-D as a separate paragraph.

Response 2-1: Thank you for these important points and suggestions. Accordingly, we moved the results of the metadata-based interpretation analysis before the 2D visualization analysis.

As the reviewer correctly described, the rationale behind the metadata-based analysis was to provide an interpretation of PCs independent from the 2D visualization analyses (Figs. 3, 4). However, according to the above change in paragraph order, we slightly modified the description of our motivation for the metadata-based interpretation. This analysis is now inserted before the 2D visualization analysis, and thus mentioning the 2D visualization in this place may be confusing to readers:

“Metadata-based interpretation of the task organizations in the cerebellum and subcortex

Although the similarity-based analyses in the previous section showed different representation patterns of 103 tasks in the cortex, cerebellum, and subcortex, it was unclear what cognitive factors contributed to these organizations. To interpret cognitive factors related to those tasks, we first performed principal component analysis (PCA) on the averaged weight matrix of the task-type model concatenated across six subjects. The resultant PCs were then associated with independent cognitive factors using a metadata-based reverse inference analysis.”

The results of 2D visualization (Fig 3a–b, 4a–b) were reported in the following subsection. We modified the subsection title to **“Visualization of representational structures of diverse tasks in the 2-dimensional cognitive spaces”** and added the following text:

“To provide a visual representation of diverse cognitive functions in different brain regions, we mapped all tasks onto 2-dimensional cognitive spaces using the loadings of the first and second PCs as the x-axis and y-axis, respectively.”

We then reported results in cerebellum and subcortex subregions in separate paragraphs, adding the following phrase about analysis motivation:

“To scrutinize representational differences in each subregion of the cerebellum and subcortex, we visualized average weight values of 103 tasks in each subregion (Fig. 3c–d, 4c–d).”

We also provided a detailed explanation about the consistency of metadata-based inference and 2D visualization as follows:

“The 2D map based on the cerebellum showed consistent task organization with the metadata-based interpretation of PCs. Introspection tasks (“ImagineFuture”, “RecallFace”) are colored in red and located on the right side (i.e., on the positive side of the PC1). Tasks related to the executive function (“CalcHard”, “PropLogic”) are colored in green and located on top (i.e., on the positive side of the PC2). Language

tasks (“Sarcasm”, “WordMeaning”) are colored in blue. The 2D map obtained based on the subcortex also showed consistent task organization with the metadata-based interpretation of PCs. Emotional tasks (“RateHappyPic”, “RateDisgustPic”) are colored in red and located on the right side (i.e., on the positive side of the PC1). Memory tasks (“LetterFluency”, “RecallFace”) are colored in green and located on top (i.e., on the positive side of the PC2). Motor tasks (“PressOrdHard”, “PressLeft”) are colored in blue.”

Please note that we mentioned only the positive sides of PCs in these phrases for the sake of clarity. We can, however, see that executive function tasks in the cerebellum (“CalcHard” and “PropLogic”) are not colored in red, consistent with the result that the negative side of PC1 was associated with executive function. Emotion-related tasks in the subcortex (“RateHappyPic” and “RateDisgustPic”) are not colored in blue, which is consistent with the result that the negative side of PC3 was associated with emotion. You may also refer to the response to Comment 2-2 for the discussion regarding the negative sides of PCs).

Comment 2-2: Another comment about the PCA interpretation analysis – are there many voxels with a negative score for one or more of the PCs? If so, how should we interpret the cognitive factors that have negative weights (“bottom” terms in Supp Table 2)? It seems like these negatively weighted cognitive factors might have a meaningful interpretation, for example voxels with a negative score on PC1 should in theory be modulated by the cognitive factors that are negatively associated with PC1. Currently it seems like there is a focus only on the “positive” side of each PC, without addressing the “negative” side, so this warrants at least some discussion.

Response 2-2: We could interpret the “negative side” in the reviewer’s comment as cognitive factors showing negative correlation coefficients. Under this interpretation, we agree with the reviewer that the negative sides of PCs are also informative. In fact, we have reported an interpretation of negative sides in a previous manuscript, such as in the **Metadata-based interpretation of the task organizations in the cerebellum and**

subcortex subsection:

“For the cerebellum, PC1 was associated with introspection and emotion terms on the positive side (top 10 terms; “theory mind,” “disgust”) and executive function and motor terms on the negative side (bottom 10 terms; “working memory,” “motor”).”

To reflect this interpretation in the main figures, we modified the color bars under Figs 3a and 4a to indicate both positive and negative sides, as well as added labels to the negative sides (e.g., “Introspection” and “Executive function” labels to the red color bar). We further discussed this topic in the response to Comment 2-3.

Comment 2-3: More discussion should be given to the differences between the cognitive task spaces encoded in the cerebellum versus subcortex and the cortex (Figs 3-4). For example, this sentence seems like an over-simplification: (line 212) “Introspection, executive function, motor function, and language are predominantly represented in the cerebellum, whereas motor function, emotion, and memory are predominantly represented in the subcortex.” What about the cognitive factors that are negatively weighted for each PC? Also, if motor function is represented strongly by both cerebellum and subcortex, it doesn’t make sense to say it is predominantly represented by either of them. Some text in the discussion should be devoted to the difference between these brain regions (and versus the cortex) and how it relates to past work.

Response 2-3: We agree with the reviewer that focusing only on the positive side of PCs was an oversimplification and that the predominance of motor functions in cerebellum and subcortex does not make sense. We thus removed the following paragraph after the metadata-based inference analysis from the revised manuscript:

~~“These results indicate that although abstract cognitive functions are represented in the cerebellum and subcortex, their distributions are distinct. Introspection, executive function, motor function, and language are predominantly represented in the cerebellum, whereas motor function, emotion, and memory are predominantly represented in the subcortex.”~~

In the Discussion section, we added the following information regarding cognitive factors represented on positive and negative sides in the cerebellum and subcortex:

“The metadata-based inference analysis revealed that both positive and negative sides of the top five PCs were associated with introspection/emotion, executive function, language, and motor terms. The involvement of the cerebellum in these cognitive factors has been reported in various previous studies (e.g., motor^{9,10}, language^{11,47}, emotion/introspection^{12,48}, and executive function^{14,49}). We further investigated the functional contributions of cerebellum subregions to these cognitive factors using the functional parcellation of King et al. (2019)¹⁷. We adopted functional ROIs (fROIs) instead of anatomical ROIs because the study reported the dissociation between anatomical and functional parcellation. In line with this, fROIs have functional labels, which would be appropriate for testing the validity of the current study. Consistent with functional labels, we found that motor tasks such as “PressRight” had a larger weight than “PressLeft” in the cerebellar subregion MDTB_2 (Right-hand presses), whereas “PressLeft” had a larger weight in the MDTB_1 (Left-hand presses) (Fig. 3c, Supplementary Fig. 5a). Language-related tasks such as “WordMeaning” and “MoralPersonal” had larger weights on the positive side of PC3 (colored blue in Fig. 3a, related to language terms) in the MDTB 7 (Narratives) and MDTB 8 (Word comprehension) (Fig. 3d, Supplementary Fig. 5f). Demanding tasks such as “CalcHard” and “RelationLogic” had larger weights on the negative side of the PC1 and positive side of the PC2 (related to executive function terms) in the MDTB-5 and MDTB-6 (Divided attention) (Supplementary Fig. 5d, 5e). Imagination and recall tasks such as “RecallFace” and “ImagineFuture” had larger weights on the positive side of PC1 (related to introspection terms) in the MDTB 10 (Autobiographical recall) (Supplementary Fig. 5h). These results confirmed the validity of functional parcellation in the cerebellum and demonstrated the diversity of task representations even within the same fROIs.”

“As for the subcortex, both positive and negative sides of the top five PCs were associated with emotion, memory, and motor terms, whereas the negative sides were further associated with somatosensory terms. The association of these terms was likely due to the contribution of subcortex subregions. The bilateral amygdala had larger

weights on the positive side of PC1 (related to emotion terms; Supplementary Fig. 6c, d), which is consistent with previous studies reporting involvement of this region in emotion recognition^{23,24}. The bilateral hippocampus was more weighted on the positive side of PC2 (related to emotion and memory terms; Fig. 4d and Supplementary Fig. 6b), which was consistent with previous studies in episodic memory and retrieval^{19,20,22}. The bilateral caudate was more weighted on the negative side of PC1 (related to motor terms; Fig. 4d and Supplementary Fig. 6b), in line with previous studies of motor control and learning^{50,51}. The bilateral thalamus was more weighted on the negative side of PC2 (related to somatosensory terms; Fig. 4d and Supplementary Fig. 6b), which is consistent with the widely-known view of this region as a pathway of sensory information^{52,53}. Note that we used anatomical ROIs in the analysis of subcortex subregions. Although a recent study provided a detailed parcellation based on functional connectivity gradients⁵⁴, we did not adopt this parcellation because the functional labels were not provided for this atlas. Further investigation may clarify distinct cognitive spaces within each subcortical structure using such parcellation.”

We added the following references:

“Van Overwalle, F., Ma, Q. & Heleven, E. The posterior crus II cerebellum is specialized for social mentalizing and emotional self-experiences: a meta-analysis. Soc. Cogn. Affect. Neurosci. 15, 905–928 (2020).

Koziol, L. F., Budding, D. E. & Chidekel, D. From movement to thought: executive function, embodied cognition, and the cerebellum. Cerebellum 11, 505–525 (2012).

Choi, Y., Shin, E. Y. & Kim, S. Spatiotemporal dissociation of fMRI activity in the caudate nucleus underlies human de novo motor skill learning. Proc. Natl. Acad. Sci. U. S. A. 117, 23886–23897 (2020).

Hardwick, R. M., Rottschy, C., Miall, R. C. & Eickhoff, S. B. A quantitative meta-analysis and review of motor learning in the human brain. Neuroimage 67, 283–297 (2013).

Sherman, S. M. Thalamus plays a central role in ongoing cortical functioning. Nat. Neurosci. 19, 533–541 (2016).

Halassa, M. M. & Sherman, S. M. Thalamocortical Circuit Motifs: A General

Framework. Neuron 103, 762–770 (2019).”

Comment 2-4: For the statistical testing on your decoding results (lines 225-245, and 644-646), some justification is needed on the choice of a one-sided sign test for evaluating significance (maybe there is a citation for this?). I am not sure this is an appropriately sensitive test for evaluating significance. A more traditional and robust method would be to use a permutation test, where you shuffle the labels of the dataset many times to generate a null distribution of accuracy, and then compare your real decoding accuracy to the null distribution to get a p value. You could do this for each of the tasks individually, or you could just compute a permuted p-value for the average decoding accuracy across all tasks (the latter might be simpler and less confusing for the reader).

Response 2-4: The sign test is a nonparametric statistical test (Hollander et al., 2013), which has been used in previous decoding studies (e.g., Pereira et al., Nat Commun 2018; Rutishauser et al., Neuron 2018; Sergent et al., Nat Commun 2021), including our previous study (Nakai & Nishimoto, Nat Commun 2020). To facilitate comparison of the present study with our previous study, which focused on the cortex, we believe that it is important to report the decoding results using the same statistical testing (i.e., sign test).

Based on the reviewer’s suggestion, we added the following references regarding the sign test:

“Hollander, Myles, Douglas A. Wolfe, and Eric Chicken. *Nonparametric Statistical Methods.pdf*. (John Wiley & Sons, 2013).

Rutishauser, U., Aflalo, T., Rosario, E. R., Pouratian, N. & Andersen, R. A.

Single-Neuron Representation of Memory Strength and Recognition Confidence in Left Human Posterior Parietal Cortex. *Neuron* 97, 209-220.e3 (2018).

Pereira, F. et al. Toward a universal decoder of linguistic meaning from brain activation.

Nat. Commun. 9, 963 (2018).

Sergent, C. et al. Bifurcation in brain dynamics reveals a signature of conscious processing independent of report. *Nat. Commun.* **12**, 1149 (2021).”

We also referred to this citation in the **Statistics and Reproducibility** subsection:

“The statistical significance of the decoding accuracy (for both one-vs-one and task score-based methods) was tested for each task using the one-sided sign test (102 samples; $P < 0.05$, with FDR correction)⁶⁴, which is a nonparametric statistical test used in previous decoding studies^{45,65–67}.”

The permutation test is a widely-used and robust method for decoding analysis. We thus additionally performed permutation tests by shuffling task labels 5,000 times. We compared the average decoding accuracy across all tasks with the null distribution of decoding accuracy by obtaining one permuted p-value for each brain region. This analysis showed that tasks were significantly decoded in all three brain regions for all subjects ($P < 0.001$).

To explain this result, we added the following text in the Results section:

“To check the robustness of our decoding results, we performed permutation tests by randomly shuffling task labels in the test dataset for a total of 5,000 times. This analysis showed that tasks were significantly decoded in all three brain regions for all subjects ($P < 0.001$).”

Moreover, we also added the following text in the **Statistics and Reproducibility** subsection:

“The significance of average decoding accuracy (across tasks) was further tested using the permutation test ($P < 0.05$). Specifically, task labels in the test dataset were randomly shuffled 5,000 times, and p -values were calculated (for both one-vs-one and task score-based methods) based on the resultant null distribution of average decoding accuracy.”

Comment 2-5: Some clarity is also needed on how/whether decoding results were combined across subjects (line 646 “Results from the six subjects were considered as

replications of the analyses.”). Does this mean that you get a p-value for every individual subject? If so, are the “percent significant tasks” in the text (line 229) just for one subject? If they are, the values for every subject should be reported somewhere, or subject-averaged values should be reported.

Response 2-5: We obtained p-values and a percent of significant tasks for each individual subject. This is why the percentage of significant tasks was reported with a mean and SD (across six subjects). Although we already reported subject-averaged values in the original manuscript, our description was not clear enough. To avoid confusion, we modified the following paragraphs in the **Decoding of novel cognitive tasks from the activity of the cortex, cerebellum, and subcortex** subsection:

“Significance of the decoded accuracy was further evaluated using a one-sided sign test and; more than 99% of tasks were significantly decoded (mean \pm SD, 99.5% \pm 0.5% of the tasks were significant; $P < 0.05$, false discovery rate [FDR]-corrected).”

“We also found that more than 90% of the cognitive tasks were significantly decoded using only cerebellar voxels (decoding accuracy, mean \pm SD, 0.918 \pm 0.015; 97.1% \pm 1.9% of the tasks were significant; Fig. 5b middle) or subcortical voxels (decoding accuracy, mean \pm SD, 0.856 \pm 0.013; 92.7% \pm 2.2% of the tasks were significant; Fig. 5b bottom)”

“The model decoded most cognitive tasks from cortical (decoding accuracy, mean \pm SD, 0.950 \pm 0.08; 99.2% \pm 0.7% of the tasks were significant)”

Based on the reviewer’s suggestion, we added Supplementary Tables 4–9 including decoding accuracy and the percent of significant tasks for each individual.

Supplementary Table 4. Mean decoding accuracy with the one-vs-one method

	ID01	ID02	ID03	ID04	ID05	ID06
	Original					
Cortex	0.965	0.951	0.952	0.953	0.937	0.955

Cerebellum	0.929	0.917	0.926	0.932	0.891	0.911
Subcortex	0.873	0.861	0.849	0.858	0.834	0.860
	After excluding sensorimotor voxels (Thresholds = 0.3)					
Cortex	0.963	0.949	0.951	0.952	0.936	0.948
Cerebellum	0.925	0.920	0.923	0.926	0.888	0.907
Subcortex	0.872	0.862	0.849	0.858	0.838	0.858

Supplementary Table 5. Percentage of significant tasks with the one-vs-one method

	ID01	ID02	ID03	ID04	ID05	ID06
	Original					
Cortex	100.0%	99.0%	100.0%	100.0%	100.0%	99.0%
Cerebellum	100.0%	97.1%	97.1%	98.1%	94.2%	96.1%
Subcortex	94.2%	93.2%	90.3%	91.3%	91.3%	96.1%
	After excluding sensorimotor voxels (Thresholds = 0.3)					
Cortex	100.0%	98.1%	99.0%	100.0%	99.0%	99.0%
Cerebellum	100.0%	97.1%	96.1%	98.1%	94.2%	95.1%
Subcortex	94.2%	93.2%	90.3%	91.3%	90.3%	95.1%

Supplementary Table 6. Mean decoding accuracy with the task score method

	ID01	ID02	ID03	ID04	ID05	ID06
	Original					
Cortex	0.615	0.597	0.591	0.594	0.560	0.595
Cerebellum	0.540	0.517	0.526	0.538	0.467	0.529
Subcortex	0.413	0.399	0.388	0.415	0.379	0.432
	After excluding sensorimotor voxels (Thresholds = 0.3)					
Cortex	0.601	0.580	0.577	0.583	0.551	0.584
Cerebellum	0.532	0.509	0.517	0.528	0.463	0.522
Subcortex	0.412	0.399	0.387	0.415	0.380	0.431

Supplementary Table 7. Percentage of significant tasks with the task score method

	ID01	ID02	ID03	ID04	ID05	ID06
	Original					
Cortex	100.0%	100.0%	99.0%	99.0%	100.0%	99.0%
Cerebellum	99.0%	99.0%	99.0%	99.0%	98.1%	98.1%
Subcortex	97.1%	97.1%	94.2%	92.2%	94.2%	98.1%
	After excluding sensorimotor voxels (Thresholds = 0.3)					
Cortex	100.0%	98.1%	99.0%	99.0%	99.0%	99.0%
Cerebellum	99.0%	99.0%	98.1%	99.0%	96.1%	98.1%
Subcortex	97.1%	97.1%	94.2%	92.2%	95.1%	98.1%

Supplementary Table 8. Mean decoding accuracy using the support vector machine

	ID01	ID02	ID03	ID04	ID05	ID06
	Original					
Cortex	98.7%	97.6%	98.1%	98.7%	93.3%	98.3%
Cerebellum	90.6%	90.5%	88.6%	90.1%	75.3%	89.7%
Subcortex	75.0%	73.7%	79.2%	80.1%	69.9%	75.8%

Supplementary Table 9. Percentage of significant tasks using the support vector machine

	ID01	ID02	ID03	ID04	ID05	ID06
	Original					
Cortex	100.0%	100.0%	100.0%	100.0%	100.0%	100.0%
Cerebellum	100.0%	100.0%	100.0%	100.0%	92.2%	100.0%
Subcortex	96.1%	95.1%	97.1%	97.1%	86.4%	98.1%

We referenced the above tables in the **Decoding of novel cognitive tasks from the activity of the cortex, cerebellum, and subcortex** subsection as follows:

“The decoding model of the cerebral cortex significantly decoded more than 95% of the cognitive tasks (decoding accuracy, mean \pm SD, 0.952 ± 0.009 ; Fig. 5b top, Supplementary Tables 4–5)”

“We found that $99.5\% \pm 0.5\%$ of tasks were significantly decoded using cortical activity (decoding accuracy, mean \pm SD, 0.592 ± 0.018 ; Fig. 5c top, Supplementary Tables 6–7)”

“(Supplementary Fig. 8, Supplementary Tables 4–5).”

“(Supplementary Fig. 11, Supplementary Tables 8–9)”

Comment 2-6: I appreciate the authors’ clarification on the point of the difference between the task score (which is a correlation coefficient) and the decoding accuracy. However I am still not following the analysis in lines 236-245. For example “we quantified decoding accuracy using task score”. How are you quantifying decoding accuracy and computing significance from the task scores alone? It seems like in order to get decoding accuracy, you need to compare the task score against the other tasks, and that would be the same as the previous analysis (lines 225-235). Are you just comparing the task score versus its chance value (0.5)? If so, this doesn’t seem like an appropriate method as this is not a robust way to evaluate significance (see notes above on permutation testing).

Response 2-6: As the reviewer correctly pointed out, we computed the significance of task score decoding accuracy by comparing task scores to the chance value (0) using the one-sided sign test. To avoid confusion, we modified the following text in the **Statistics and Reproducibility** subsection:

“The statistical significance of the decoding accuracy (for both one-vs-one and task score-based methods) was tested for each task using the one-sided sign test (102 samples; $P < 0.05$, with FDR correction).”

We added the following text in the Fig. 5 legend:

“c Histogram of task decoding accuracies represented by average task score. The red line indicates the chance-level accuracy (0).”

We also evaluated significance using a permutation test like in Response 2-4 and found that tasks were significantly decoded in all three brain regions for all subjects ($P < 0.001$). To explain this result, we added the following text in **Decoding of novel cognitive tasks from the activity of the cortex, cerebellum, and subcortex** subsection:

“Permutation tests also showed that tasks were significantly decoded in all three brain regions for all subjects ($P < 0.001$).”

Comment 2-7: The interpretation of the difference in standard deviation between the correlation coefficients in the different brain areas needs more justification (Fig 2f, line 101) “...suggesting that the structure of cognitive tasks is more distinctively organized in the cortex compared to the cerebellum and subcortex”. The fact that the standard deviation is smaller for the cerebellum and subcortex than the cortex doesn’t necessarily mean that there is more “distinctiveness” among the different tasks. It seems like the cortex has more large positive correlations between tasks than the other brain areas (for example in Fig 2d-e), which should mean that tasks are more similar there, not less distinctive? The difference in the distribution of correlation coefficients between weights across the different brain regions could also be due to differences in noise/signal quality across brain regions. These issues should be discussed.

Response 2-7: Thank you for the important indication, we separately answered the two points raised by the reviewer.

As for the first point (“It seems like the cortex has more large positive correlations between tasks than the other brain areas”), there are task pairs that are more similar in the cortex, but there are also those that are *less* similar in the cortex. In fact, 49.3% of task pairs have a greater similarity in the cortex than in the cerebellum, while 50.7% of task pairs have a lesser similarity in the cortex than in the cerebellum.

Moreover, 48.7% of task pairs have greater similarity in the cortex than in the subcortex, while 51.3% of task pairs have lesser similarity in the cortex than in the subcortex. The horizontally-spread distributions in Fig. 2d–e reflect these patterns (i.e., spread in both positive and negative directions on the x-axis).

Regarding the second point (“The difference ... could also be due to differences in noise/signal quality across brain regions.”), we completely agree with the reviewer that the difference in distribution of weight correlations across the different brain regions could be due to the difference in signal/noise quality. It is possible that task representations are equally distinct in the cerebellum and subcortex but are less clear due to the signal/noise quality of the current fMRI measurement. We note, however, that the signal/noise quality issue does not affect our main conclusion that complex cognitive functions are represented in the cerebellum and subcortex.

Regarding the second point, we added the following text in the Discussion section:

“Although the RSM showed a smaller SD in the cerebellum and subcortex than in the cortex, such difference in task similarity might be caused by the low signal-to-noise ratio (SNR) in the cerebellum and subcortex, rather than by the intrinsic distinctiveness of task representations. It is possible that task representations are equally distinct in the cerebellum and subcortex but are less clear due to the signal/noise quality of the current fMRI measurement. Further improvement in the measurement of cerebellar and subcortical activity is needed to disentangle the SNR effect from the distinctiveness of task representations across different brain regions.”

Comment 2-8: Since the results text comes before methods here, it would be helpful to provide brief description of some of the major methods when they are first introduced in the text, or at least provide a parenthetical reference to the relevant section of the methods. For example, when you first mention the “sparse task type” model on line 88, you could have a brief note describing what the features of that model are so that the reader doesn’t have to flip all the way to the methods section. Same can be done for the “latent cognitive factors” (line 221), and the “sensorimotor features” first mentioned at line 107.

Response 2-8: We agree with the reviewer that describing methodology in the Results section will help readers to understand our results. We added the following paragraphs in the **Brain representations of task structures were preserved across the cortex, cerebellum, and subcortex** subsection:

“..., we constructed a series of encoding models using sparse task-type features composed of one-hot vectors corresponding to the 103 tasks.”

“The similarity of task structures across the three brain parts could be obtained merely by the difference of task-specific sensorimotor information. To exclude such a possibility, we first extracted visual features using the motion energy (ME) model, auditory features using the modulation transfer function (MTF) model, and motor features using the button response (BR) model (see **Motion energy features, Modulation transfer function features, and Button response features** subsections in Methods for detail) and concatenated those features to obtain sensorimotor features. We then performed the encoding model analyses 50 times within the training dataset using the sensorimotor features and excluded the reliably predicted voxels...”

In the **Decoding of novel cognitive tasks from the activity of the cortex, cerebellum, and subcortex** subsection:

“Cognitive factor features were calculated based on Pearson’s correlation coefficients between the weight maps of the task-type model and reverse inference maps of the Neurosynth⁴⁶ (see the **Cognitive factor features** subsection in Methods for details).”

Furthermore, we added the following descriptions of analysis details for clarity:

In the **Brain representations of task structures were preserved across the cortex, cerebellum, and subcortex** subsection:

“Therefore, we visualized the representational structures of cognitive tasks using the representational similarity matrix (RSM) based on the weight matrices of the task-type encoding models for each region (Fig. 2a–c) concatenated across six subjects. The RSM was obtained by calculating the Pearson’s correlation coefficients between the averaged weights of all task pairs across three time delays, concatenated across six subjects.”

In the **Visualization of representational structures of diverse tasks in the 2-dimensional cognitive spaces** subsection:

“we mapped all tasks onto 2-dimensional cognitive spaces using the loadings of the first and second PCs as the x-axis and y-axis, respectively.”

Comment 2-9: In Figures 3 and 4, there should be a separate legend for panels C-D which indicate what the red/blue colors and dot sizes mean. Especially because the red/blue colors are also used in panel A, it looks as if the red/blue in the legend of A are referring to C-D. You could also consider changing to colors other than red/blue to help with the interpretability.

Response 2-9: Based on the reviewer’s suggestion, we changed the circle colors in Figure 3c–d, Figure 4c–d, as well as Supplementary Figures 4 and 5 into yellow and cyan. We also clarified that circle colors refer to panels c and d by adding the following text in the legends of Figures 3 and 4:

“Tasks with positive and negative weight values in c and d were indicated in yellow and cyan, respectively.”

We also changed the following text in the legends of Supplementary Figures 4 and 5:

“Tasks with positive and negative weight values were denoted in yellow and cyan, respectively.”

Comment 2-10: Line 161 - Supplementary Table 1 should be Supplementary Table 2

Response 2-10: Thank you for pointing out this mistake. We also found that Supplementary Table 2 in the Fig. 4 legend should be Supplementary Table 3. We thus modified the following sentences in each legend:

“Each PC is labeled based on metadata-based interpretation analysis (Supplementary Table 2).”

“Each PC is labeled based on metadata-based interpretation analysis (Supplementary Table 3).”

Comment 2-11: Line 622 - “We then calculated the time-averaged task scores for each task using the one-vs.-one method” – this is confusing, is this supposed to say something like “We then calculated the time-averaged task scores for each task, and then performed decoding using the one-vs.-one method.” ?

Response 2-11: We apologize for the lack of clarity. We have modified the sentence as suggested:

“We then calculated the time-averaged task scores for each task, and then performed decoding using the one-vs.-one method.”

Comment 2-12: Figure 6 - Some of the text is cut off in the titles of each panel, make sure to check this

Response 2-12: Thank you for pointing out our mistakes. We corrected the title positions of panels b, c, and d in Fig. 6.

Comment 2-13: Fig 6f – are error bars across subjects, or across voxels? Please clarify.

Response 2-13: Error bars were calculated across subjects. We added the following text in the Fig. 6 legend:

“Error bar, SD (calculated across subjects).”

In addition to the above changes made for the responses to Reviewer's comments, we also made the following changes:

- (1) In the previous version, we labeled PC5 of the cerebellum with "cognitive demand," but we think that it would be less confusing to readers to label it with "executive function."
- (2) We recognized that the percent description of decoding accuracy in one-vs-one method is confusing and is inconsistent with descriptions in Fig. 5. We thus stopped using percent descriptions for the decoding accuracy of the one-vs-one method (e.g., "95.2% => 0.952").

REVIEWERS' COMMENTS:

Reviewer #1 (Remarks to the Author):

I have no more comments.

Reviewer #2 (Remarks to the Author):

The authors have done a great job addressing my concerns with the manuscript, and I appreciate their thorough and detailed responses. I do not see any other concerns with this manuscript at this point.

One minor comment: In figure 6, it still looks like a lot of the text labels are cut off (even though in the rebuttal the authors said they have fixed this). Maybe it is an issue with PDF conversion? Just double check this in the final version.

We greatly appreciate the reviewers' time and effort in reviewing our manuscript and providing helpful comments. Based on their comments, we have revised the original manuscript. Please find our responses to the comments below, along with the related changes in the revised manuscript.

Reviewer #2 (Remarks to the Author):

Comment 2-1: One minor comment: In figure 6, it still looks like a lot of the text labels are cut off (even though in the rebuttal the authors said they have fixed this). Maybe it is an issue with PDF conversion? Just double check this in the final version.

Response 2-1: Thank you for this indication. The text cut-off might be caused by the pdf conversion because those texts are not cut off in our original figure and word file. We checked the figure quality again and confirmed that no cut-off issue was found when the figures were uploaded as separate files (not within the word file).

In addition to the above changes made for the responses to Reviewer's comments, we also made the following changes, which is mostly based on the Final Revision Instruction document:

Based on the suggestion by the editor, we changed the title as follows:

“Representations and decodability of diverse cognitive functions are preserved across the human cortex, cerebellum, and subcortex”

We rewrote our abstract in the present tense where possible.

We replaced original pictures used in Figure 1a with those obtained from StoryBlock (<https://www.storyblocks.com/>), which provides Royalty-free (re-distributable) movie stimuli. We thus think that there is no copy right problem.

We added sample size information in legends: “(n = 5,253)” in Fig. 2 legend. “(n = 715)” and “(n = 103)” in Fig. 5 legend, “n = 412” and “n = 6” in Fig. 6 legend.

We added Supplementary Data 1-5 and added the following text in **Data availability** section:

“Source data underlying Figs. 2d-f, 3b, 4b, 5b-e, and 6f are provided in Supplementary Data 1, 2, 3, 4, and 5, respectively. Other data are available from the corresponding author upon reasonable request.”

While preparing these data, we noticed that the Figure panel 2f was an old version, which contained data with a script mistake. We thus replaced this panel with the new version. Note that this change does not have any impact on the interpretation and conclusion of our study (both versions show bar graphs of Cortex > Cerebellum > Subcortex) and does not cause any changes in the main text.

We added the following citation of the dataset:

“Nakai, T., & Nishimoto, S. Over 100 Task fMRI Dataset. OpenNeuro. [Dataset] doi: 10.18112/openneuro.ds002306.v1.0.3. (2020)”

This citation was referred in Data availability section:

“The raw MRI data are available at the OpenNeuro.org (<https://openneuro.org/datasets/ds002306>)⁶⁹.”